# Understanding Sparse Neural Networks from their Topology via Multipartite Graph Representations

**Elia Cunegatti**                                                                     *elia.cunegatti@unitn.it*
*University of Trento, Italy*

**Matteo Farina**                                                                          *m.farina@unitn.it*
*University of Trento, Italy*

**Doina Bucur**                                                                          *d.bucur@utwente.nl*
*University of Twente, The Netherlands*

**Giovanni Iacca**                                                                  *giovanni.iacca@unitn.it*
*University of Trento, Italy*

**Reviewed on OpenReview:** *https://openreview.net/forum?id=Egb0tUZnOY*

## Abstract

Pruning-at-Initialization (PaI) algorithms provide Sparse Neural Networks (SNNs) which are computationally more efficient than their dense counterparts, and try to avoid performance degradation. While much emphasis has been directed towards *how* to prune, we still do not know *what topological metrics* of the SNNs characterize *good performance*. From prior work, we have layer-wise topological metrics by which SNN performance can be predicted: the Ramanujan-based metrics. To exploit these metrics, proper ways to represent network layers via Graph Encodings (GEs) are needed, with Bipartite Graph Encodings (BGEs) being the *de-facto* standard at the current stage. Nevertheless, existing BGEs neglect the impact of the inputs, and do not characterize the SNN in an end-to-end manner. Additionally, thanks to a thorough study of the Ramanujan-based metrics, we discover that they are only as good as the *layer-wise density* as performance predictors, when paired with BGEs. To close both gaps, we design a comprehensive topological analysis for SNNs with both linear and convolutional layers, via (i) a new input-aware Multipartite Graph Encoding (MGE) for SNNs and (ii) the design of new end-to-end topological metrics over the MGE. With these novelties, we show the following: (a) The proposed MGE allows to extract topological metrics that are much better predictors of the accuracy drop than metrics computed from current input-agnostic BGEs; (b) Which metrics are important at different sparsity levels and for different architectures; (c) A mixture of our topological metrics can rank PaI algorithms more effectively than Ramanujan-based metrics. The codebase is publicly available at https://github.com/eliacunegatti/mge-snn.

## 1 Introduction

Pruning Dense Neural Networks (DNNs) has recently become one of the most promising research areas in machine learning. Pruning removes a portion of network parameters (*i.e.*, weights) to reduce the computational resources and inference time, while avoiding performance degradation. The *"winning tickets"* hypothesis, a milestone discovery from Frankle & Carbin (2019), states that within randomly initialized DNN there exist subnetworks that can reach the performance of the overall DNN when trained in isolation. Thanks to this, among several alternatives, Pruning at Initialization (PaI) emerged as a good pruning approach.

While many papers have investigated *how* to find sparse architectures, only a few looked at *what characterizes a well-performing SNN* from a topological viewpoint. An SNN is characterized by a unique topology, and many strategies can be employed to map this topology to a graph representation, allowing to conduct various

kinds of analyses (Pal et al., 2022; Hoang et al., 2023b;a). For Convolutional Neural Networks (CNNs), state-of-the-art Graph Encodings (GEs) of SNNs rely on layer-based *rolled* representations, where nodes map to layer parameters, *i.e.*, either a kernel channel (Prabhu et al., 2018), or a kernel value (Pal et al., 2022; Hoang et al., 2023b). While highly informative, these studies neglect the impact of the *input dimensionality* on the graph topology, and overlook the importance of *end-to-end* information processing within neural networks, solely employing layer-based Bipartite Graph Encodings (BGEs).

To overcome both limitations, we propose a novel *unrolled input-aware* Multipartite Graph Encoding (MGE) which fully captures the end-to-end relationship between the (pruned) network parameters and the input dimensionality. Given an SNN and the dimensionality of its input, our GE works in two steps. First, we generate a BGE for each layer, where left/right nodes correspond to the unrolled inputs/outputs (*e.g.*, pixels in feature maps) while edges correspond to the masked/unmasked relations between them, according to the pruned layer parameters. Since we anchor the dimensionality of our BGEs to the layers' inputs and outputs, with this strategy we generate consecutive bipartite graphs where the right nodes of layer $l$ correspond to the left nodes of layer $l + 1$. This enables extending our sequence of layer-based BGEs to a unique MGE, which fully captures the SNN in an end-to-end and data-aware manner.

We then analyze the accuracy drop of SNNs differently from previous works. Within the field, most efforts are currently aimed at understanding the relationship between SNN-derived GEs and *Ramanujan graphs*, a special class of graphs based on the theory of Expanders, *i.e.*, graphs with high connectivity but few edges (Pal et al., 2022; Hoang et al., 2023b;a). We discover that the existing Ramanujan-based metrics often do not provide more information than a readily available quantity after pruning: the average *layer-wise* density. This finding has impact, since Ramanujan-based metrics require the cumbersome process of building BGEs, while the average layer-wise density can be computed directly from the pruned weight matrices (or convolutional kernels) of the SNN. Hence, in this work we analyze SNNs from a broader *graph theory* perspective, and propose a different set of graph-based metrics for this purpose. The proposed MGE, paired with the proposed metrics, turns out to be crucial for the analysis of SNNs, underlining the importance of SNN-derived GEs.

**Contributions.** To summarize, the main contributions of this paper can be outlined as follows:

(a) We examine the link between the Ramanujan-based metrics commonly used for graph-based SNN analysis and the average layer-wise density, finding that the former are seldom more informative than the latter.

(b) We present a new *unrolled input-aware* MGE that fully represents the end-to-end topology of an SNN while accounting for its input dimensionality.

(c) We shift the perspective of graph-based SNN analysis from *Ramanujan-based* to a broader viewpoint grounded in *graph theory*, by proposing new topological metrics for SNNs.

(d) We experiment with a large pool of $1,260$ SNNs, showing that the proposed MGE and metrics successfully enable *predicting the accuracy drop* of SNNs, and *ranking* PaI methods according to their expected performance, thus paving the way towards the adoption of SNN-derived GEs in practice. We discover that no metric alone can fully explain the performance drop of SNNs for every sparsity ratio and architecture setting, rather a combination of metrics is needed to understand the reason behind the performance drop of SNNs in an arbitrary scenario.

## 2 Related Work

In this Section, we first introduce the Pruning at Initialization algorithms, and then discuss how graph theory intersects with Deep Learning, specifically with SNN analysis.

**Pruning at Initialization (PaI).** This family of pruning algorithms aims at discovering the best-performing subnetwork by removing weights *prior to training*. Good solutions to PaI are very appealing, thanks to their potential to decrease the burden of dense optimization. In modern literature, the earliest approach of this kind is SNIP (Lee et al., 2019), which selects the connections to preserve based on their estimated influence on the loss function. While SNIP works in one-shot, its iterative version IterSNIP is presented in de Jorge et al. (2020). Using second-order information, GraSP (Wang et al., 2020) applies a gradient signal preservation mechanism based on the Hessian-gradient product. Differently from data-dependent methods,

SynFlow (Tanaka et al., 2020) prunes according to the *synaptic strengths* in a data-agnostic setting, which was also explored later in PHEW (Patil & Dovrolis, 2021). More recently, ProsPR (Alizadeh et al., 2022) extended SNIP by maximizing the trainability throughout meta-gradients on the first steps of the optimization process. Lastly, NTK-SAP (Wang et al., 2023) relies on the neural tangent kernel theory to remove the least informative connections in both a weight- and data-agnostic manner. These approaches are relatively cheap in terms of computational resources, since the mask is found before training in, at most, hundreds of iterations. However, as the sparsity requirement increases, performance deteriorates faster with PaI methods than it does when other categories of pruning algorithms are employed (Dynamic Sparse Training methods above all, *e.g.*, Dettmers & Zettlemoyer (2019); Evci et al. (2020); Liu et al. (2021b)), due to the difficulty of training SNNs from scratch (Evci et al., 2019; 2022). Moreover, the ability of PaI algorithms to uncover the most effective sparse architectures has been firstly questioned using Sanity Checks in Su et al. (2020); Frankle et al. (2021). Subsequently, Liu et al. (2022) showed that small perturbations to random pruning methods, such as ER (Mocanu et al., 2017) and ERK (Evci et al., 2020), can even outperform well-engineered PaI algorithms. Consequently, this motivates us to shed light on what are the characteristics of a well-performing SNN after PaI.

**Graph Representation of SNNs and DNNs.** Studying DNNs from the perspective of graph theory is a popular approach that commonly leverages weighted graph representations. Based on these, many advances to Deep Learning have been proposed, *e.g.*, early-stopping criteria (Rieck et al., 2019), customized initialization techniques (Limnios et al., 2021), and performance predictors (Vahedian et al., 2021; 2022).

Other aspects of graph theory have been shown to be successful also in the context of SNNs. In fact, the sparse topology can explain or hint at *why* the network can still solve a given task with a minor accuracy drop w.r.t. its dense counterpart. Within the field, the random generation of small sparse structures for Multi-Layer Perceptrons (MLPs) has been proposed in Bourely et al. (2017); Stier & Granitzer (2019) and their performance investigated via basic graph properties in Stier et al. (2022). Always with MLPs, the Graph-Edit-Distance has been introduced as a similarity measure in Liu et al. (2021a), to show how the graph topology evolves during Dynamic Sparse Training algorithms, such as SET (Mocanu et al., 2017).

Shifting towards more recent and deep architectures (*i.e.*, belonging to the Resnet family), You et al. (2020) used relation graphs to show that the performance of randomly generated SNNs is associated with the *clustering coefficient* (*i.e.*, the capacity of nodes to "cluster together") and the *average path length* (*i.e.*, the average number of connections across all shortest paths) of its graph representation.

More recently, to study SNNs generated by PaI algorithms, a new line of research based on Expander graphs has emerged. Exploiting Ramanujan graphs, this research line indicates that the performance of an SNN correlates with graph connectivity (Prabhu et al., 2018; Pal et al., 2022) and spectral information (Hoang et al., 2023a). Additionally, this line suggests that such performance can be *predicted* thanks to well-designed metrics, such as the iterative mean difference of bound (IMDB) (Hoang et al., 2023b). Based on the theory of Ramanujan graphs, a few works further attempted to generate efficient SNNs (Stewart et al., 2023; Laenen, 2023). As a final note, SNN connectivity has also been studied with approximations of the computational graph, to analyze how the data in input to a given sparse structure influence the "effective" sparsity (*i.e.*, the fraction of inactivated connections), "effective" nodes (*i.e.*, nodes with both input and output connections), and "effective" paths (*i.e.*, connections connecting input and output layers) )(Vysogorets & Kempe, 2023; Pham et al., 2023)[1].

In all the aforementioned works analyzing PaI algorithms, convolutional layers are modeled either with a *rolled* GE based on the kernel parameters ($K$) (Pal et al., 2022; Hoang et al., 2023b), or with a *rolled-channel* encoding (Prabhu et al., 2018; Vahedian et al., 2022) depending on the number of input/output channels ($C_{in}$ and $C_{out}$, respectively). In both cases, the relationship between the SNNs and the input data is not encoded. On the one hand, the *rolled* GE generates a graph $G \in \mathbb{R}^{|\mathcal{L}| \times |\mathcal{R}|}$ for each layer, *s.t.*

---

[1]Of note, in Vysogorets & Kempe (2023); Pham et al. (2023) the authors used an end-to-end computational graph to represent SNNs. Their proposed metrics (effective sparsity, nodes, and paths) are based on the $l_1$ path norm computed using ones as input values over the computational graph. To note that by definition a computational graph represents mathematical operations where nodes and edges respectively correspond to operations and data between operations, while in our proposed multipartite graph nodes correspond to neurons of the SNN and an edge represents the connection between neurons, that is present if the connection is not masked in the given sparse structure.

$|\mathcal{L}| = C_{in} \cdot K_w \cdot K_h$ and $|\mathcal{R}| = C_{out}$, with a resulting number of graph edges $|E| = |W|$. This GE reflects the topology of a single layer (each non-pruned weight corresponds to an edge), but cannot be used to encode connectivity between consecutive layers, since $|\mathcal{R}_i| \neq |\mathcal{L}_{i+1}|$. On the other hand, the *rolled-channel* encoding works with layer-based graphs $G \in \mathbb{R}^{|\mathcal{L}| \times |\mathcal{R}|}$, where $|\mathcal{L}| = C_{in}$ and $|\mathcal{R}| = C_{out}$, leading to $|E| = \ker(W) \setminus \{0\}$. Here, nodes correspond to channels. By design, for any input-output channel pair, an edge exists if and only if the corresponding kernel channel is not fully pruned.

Unlike these rolled GEs, we employ an *unrolled input-aware* MGE, which we describe in the next Section.

## 3    Methodology

In this Section, we first introduce the novel *unrolled input-aware* GE and its formulation in the *bipartite* version. We then show how this is extended to a *multipartite* version, which links consecutive layers and generates a single graph that is able to represent the SNNs **end-to-end**. Finally, we propose a set of topological metrics that can be extracted from the proposed encoding. Our notation is summarized in Table 1.

### 3.1    Bipartite Graph Encoding (BGE)

The proposed BGE, which takes into consideration the application of the convolutional steps over the inputs, encodes a neural network with $N$ layers as a list of weighted, directed, acyclic bipartite graphs $G = (G_1, \ldots, G_N)$. Due to its design, the bipartite graph construction differs for linear and convolutional layers.

**Linear Layers.**  For linear layers, we use the encoding proposed in (Rieck et al., 2019; Filan et al., 2021; Prabhu et al., 2018; Vahedian et al., 2022; Hoang et al., 2023b): denoting with $L_i$ and $R_i$ respectively the left and right layer of the $i$-th bipartite graph, and given a binary mask $M_i \in \{0,1\}^{|L_i| \times |R_i|}$, its corresponding GE is $G_i = (L_i \cup R_i, E_i)$, where $E_i$ is the set of edges present in $M_i$, *i.e.*, $(a,b) \in E_i \iff M_i^{a,b} \neq 0$, where $a \in L_i$ and $b \in R_i$.

**Convolutional Layers.**  For convolutional layers, our approach is substantially different from all prior work. Specifically, we devise our encoding based on the *unrolled* input size: given as input, for each $i$-th layer, a set of feature maps $\mathcal{I}_i \in \mathbb{R}^{h_i \times w_i \times C_{in}}$, we construct the corresponding bipartite graph as $G_i = (L_i \cup R_i, E_i)$, where again $L_i$ and $R_i$ are the two layers of the bipartite graph, and $L_i$ corresponds to the

Table 1: Notation used in the paper. Note that we focus on SNNs for vision tasks.

| Symbol | Definition |
|---|---|
| $G = (L \cup R, E)$ | bipartite graph with left node set $L$, right node set $R$ (for a total of $|L| + |R|$ nodes), and edge set $E$ |
| $N$ | number of layers |
| $\mathcal{I}_i$ | input size of $i$-th layer |
| $h_i, w_i$ | height, width of input feature map of $i$-th layer |
| $u_i, v_i$ | row, column indexes of input feature map of $i$-th layer |
| $a, b$ | nodes of $G$ with $a \in L$ and $b \in R$ |
| $M$ | binary mask of pruned/unpruned weights |
| $M_i$ | binary mask of pruned/unpruned weights of $i$-th layer |
| $W$ | model parameters after pruning |
| $W_i$ | model parameters after pruning of $i$-th layer |
| $\theta$ | model parameters |
| $h_{ker}, w_{ker}$ | height and width of kernel |
| $C_{in}, C_{out}$ | number of input and output channels |
| $P, S, f$ | padding, stride, filters |

flattened representation of the inputs [2]. The size of the layer $R_i$, *i.e.*, the output feature map, is calculated based on the input size $\mathcal{I}_i$ and the parameters $W_i \in \mathbb{R}^{c_{out} \times c_{in} \times h_{ker} \times w_{ker}}$ of the $i$-th layer:

$$|L_i| = h_i \times w_i \times c_{in} \quad |R_i| = \left(\frac{h_i - h_{ker}}{S} + 1\right) \times \left(\frac{w_i - w_{ker}}{S} + 1\right) \times c_{out}. \tag{1}$$

Differently from the linear layer case, the set of edges $E_i$ cannot be directly computed from the convolutional mask $M_i \in \{0,1\}^{c_{out} \times c_{in} \times h_{ker} \times w_{ker}}$ since the latter is dynamically computed over the input data[3]:

$$x_{u_i,v_i}^{out} = \sum_{in=0}^{c_{in}-1} \sum_{u=-h_{ker}}^{h_{ker}} \sum_{v=-w_{ker}}^{w_{ker}} I_{u_i,v_i}^{in} \times M^{out,in,u_i+u_{i+1},v_i+v_{i+1}} \quad \forall\, out \in [0, c_{out}). \tag{2}$$

---

[2]Padding nodes are included.
[3]The formula uses cross-correlation.

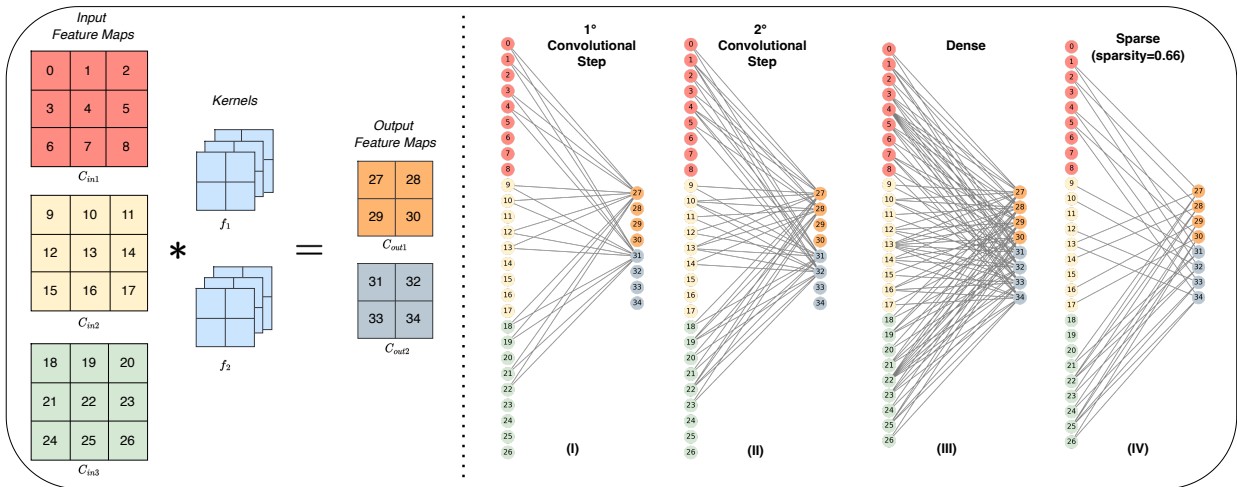

Figure 1: Illustration of the proposed unrolled input-aware BGE with $\mathcal{I} = 3 \times 3 \times 3$ and convolutional parameters ($C_{out} = 2, C_{in} = 3, w_{ker} = 2, h_{ker} = 2, P = 0, S = 1$). (I) and (II) show, respectively, the first and second convolutional steps and how the graph edges are generated assuming that all the kernel parameters are unmasked. (III) and (IV), respectively, show the complete graph representation after all the convolutional steps have been done in both the (III) dense and (IV) sparse cases.

From Eq. (1), we know that $I^{in}_{u_i,v_i}$ and $x^{out}_{u_{i+1},v_{i+1}}$ (*i.e.* the output node of the convolutional step operation) respectively correspond to a node $a_{(u_i+v_i)\times in} \in L_i$ and a node $b_{(u_{i+1}+v_{i+1})\times out} \in R_i$, so, in this case, the edges of the bipartite graph are constructed during the convolutional operation such that:

$$E_i = \{(a_{(u_i+v_i)\times in}, b_{(u_{i+1}+v_{i+1})\times out}) \mid M^{out,in,u_i+u_{i+1},v_i+v_{i+1}} \neq 0 \ \forall \, out, in, u_i, v_i\} \tag{3}$$

where the ranges of $out, in, u_i, v_i$ are defined according to Eq. (2), $in$ and $out$ denote, respectively, the input and the output channel for that convolutional step, and $\langle u_i + u_{i+1}, v_i + v_{i+1} \rangle$ corresponds to one kernel entry. Intuitively, given the $i$-th layer of a neural network layer, each input element (*e.g.*, in computer vision tasks, each pixel) represents a node in the graph, and the connection between an element of the input (denoted as $a$) and an element of the output feature map (denoted as $b$) is present if and only if during the convolutional operation, the contribution of $a$ for generating $b$ is not set to zero by the mask $M_i$ in the kernel cell used to convolute the two pixels. An illustration of such encoding, which highlights the construction of the graph throughout the convolutional steps for both SNNs and DNNs, is shown in Figure 1.

Equation 3 shows how the edges are added to the graph representation for its unweighted version. However, this equation can be easily extended to generate a weighted graph representation that does not only consider the binary mask $M$ but also the values of model parameters after pruning $W$, computed as $W = \theta \odot M$ then, in Equation 4 it is possible to see how the weighted edge list is generated.

$$E_i = \{(a_{(u_i+v_i)\times in}, b_{(u_{i+1}+v_{i+1})\times out}, W^{out,in,u_i+u_{i+1},v_i+v_{i+1}}) \mid W^{out,in,u_i+u_{i+1},v_i+v_{i+1}} \neq 0 \ \forall \, out, in, u_i, v_i\}. \tag{4}$$

## 3.2 Multipartite Graph Encoding (MGE)

The BGE described above has been devised to encode, independently, every layer (either convolutional or linear) in a network. However, the common limitation of BGEs (Pal et al., 2022; Hoang et al., 2023b) lies in the lack of connections between any two consecutive (and, indirectly, non-consecutive) $i$-th and $i + 1$-th layers. On the other hand, our proposed *unrolled input-aware* GE, differently from the existing encodings, generates by design consecutive layers with the same number of nodes between the $i$-th right layer and the $i + 1$-th left layer (*i.e.*, $|R_i| = |L_{i+1}|$) of consecutive bipartite graphs.

A multipartite graph, a.k.a. *N*-partite where $N$ represents the number of layers, is defined as $\hat{G} = (\hat{V}, \hat{E})$ where $\hat{V} = \hat{V}_1 \cup \hat{V}_2 \cup \ldots \cup \hat{V}_k$ and $\hat{E} \subseteq \{(a, b) \mid a \in \hat{V}_i, b \in \hat{V}_j, i < j, 1 \leq i, j \leq N\}$. In the previous Section,

we showed how to generate a set of bipartite graphs $G = (G_1, \ldots, G_N)$ for a network with $N$ layers where $G_i = (L_i \cup R_i, E_i)$. Given a set of bipartite graphs $G$, it is possible to concatenate all its elements into a single *multipartite* graph $\hat{G} = (\hat{V}, \hat{E})$. This concatenation holds only because with our proposed GE by design $|\hat{V}_{i+1}| = |R_i|$, hence edges can be easily rewired as: $\hat{E} = \{(a, b) \mid E_i, 1 \leq i \leq N\}$ where $a \in \hat{V}_i$ (*i.e.*, it belongs to $L_i$) and $b \in \hat{V}_{i+1}$ (*i.e.*, it belongs to $R_i$).

However, an extension of the previous encoding is needed for connecting consecutive layers when a pooling operation is employed between them. Because pooling operations lead consecutive BGEs to have different dimensions (*i.e.*, $|R_i| \neq |L_{i+1}|$), a further step is required by the GE to make such layers match in size to allow concatenation. In a nutshell, $L_{i+1}$ is reduced to $L'_{i+1}$ such that $G_{i+1} = (L'_{i+1} \cup R_{i+1}, E'_{i+1})$ and $|L'_{i+1}| = |R_i|$. The edge rewiring process to translate $E_{i+1}$ to $E'_{i+1}$ adds to $L'_{i+1}$, a total of $|\rho(R_i)|$ times, each edge in $G_{i+1}$ that links $u \in L_{i+1}$ to $v \in R_{i+1}$. This corresponds to each node in the pooling window $p_w \subseteq R_i$, computed over $R_i$, which ensures that $u \in L'_{i+1}$ is linked to $v \in R_{i+1}$. Consequently, this pooling encoding enables any pair of consecutive layers $G_i$ and $G_{i+1}$ to be linked in the MGE framework. We refer to Appendix C for the complete formal explanation of the pooling encoding. It is also worth noting that this MGE also includes residual connections, which are a crucial element in many DNN architectures, while in the previous graph encodings (Pal et al., 2022; Hoang et al., 2023a) they were analyzed layer-wise without linking them between the two corresponding layers. The concatenation of residual connections is straightforward. Given two consecutive layers linked by residual connections (e.g., the last and the first layer of the blocks in Resnet architectures), modeled by their corresponding BGEs $G_i$ and $G_{i+1}$, the residual connection is modeled as a separate bipartite graph $G^{res}_{i,i+1}$. By construction, $|L^{res}_{i,i+1}| = |L_i|$ and $|R^{res}_{i,i+1}| = |R_{i+1}|$, hence the residual connection edge construction only adds the edges of $G^{res}_{i,i+1}$ to the concatenation of $G_i$ and $G_{i+1}$, i.e., between layers $L_i$ and $R_{i+1}$.

### 3.3 Topological Metrics

The proposed unrolled MGE allows for the study of SNNs from a topological perspective, including a first analysis of the network connectivity between consecutive layers. To this aim, we define a set of *topological metrics* (in the following, referred to as *topometrics*) over SNNs, summarized in Table 2.

**Local Connectivity.** The Local metrics are graph metrics computable over individual nodes or edges. These metrics (i) are computationally inexpensive, and (ii) are able to capture some features of the graph connectivity between consecutive layers. Node-based topometrics include ① the number of *sink* nodes, ② the number of *source* nodes, and ③ the number of *disconnected nodes* over the MGE. The sink and source nodes are, respectively, those with outdegree and indegree of zero[4]. The disconnected nodes are those with neither incoming nor outgoing connections. Con-

Table 2: Proposed topometrics with their brief definitions, and their complexity ($N$: number of nodes, $E$: number of edges)

| Category | Symbol | Definition | Complexity |
|---|---|---|---|
| Local Connectivity | sink | no. of nodes with zero outgoing connections | $O(E)$ for all local metrics |
| | source | no. of nodes with zero incoming connections | |
| | disc. | no. of nodes with zero connections | |
| | r-out | no. removable outgoing connections | |
| | r-in | no. removable incoming connections | |
| Neighbor Connectivity | $\mathcal{N}_1$ | avg. number of 1-hop neighbors | $O(E)$ |
| | $\mathcal{N}_2$ | avg. number of 2-hop neighbors | $O(NE)$ |
| | $motif_{(4)}$ | no. of motifs of size $k = 4$ | $O(N3^k)$ |
| Strength Connectivity | $k$-core | centrality measure on maximal degree of subgraphs | $O(E)$ |
| | $\mathcal{S}$ | avg. nodes strength over input edges weights | $O(N + E)$ |
| Global Connectivity | $\mathcal{C}$ | no. of connected components | $O(N + E)$ |
| | $\mathcal{C}_{avg}$ | avg. size of the connected components | $O(1)$ |
| | cut-edges | no. edges to disconnected connected components | $O(N + E)$ |
| | cut-nodes | no. nodes to disconnected connected components | $O(N + E)$ |
| Expansion | $\gamma(\mathcal{A})$ | Spectral Gap of the Adjacency Matrix | $O(knm)$ [‡] |
| | $\rho(\mathcal{L})$ | Spectral Radius of the Laplacian Matrix | |

sidering the sink and source nodes, it is possible to compute the number of *removable (in/out) connections*, which are edge-based topometrics. ④ The removable out-connections of the set of source nodes (denoted here as $\alpha$) are r-out $= \frac{1}{|E|} \times \sum_{n \in \alpha}$ outdegree$(n)$. Complementary, ⑤ the removable in-connections of the set of sink nodes (denoted here as $\beta$) are r-in $= \frac{1}{|E|} \times \sum_{n \in \beta}$ indegree$(n)$. Intuitively, both these types of connections are useless for the performance of the neural network, since they are ignored at inference. Usually, lower values of Local Connectivity metrics are related to highly connected structures.

---

[‡]The complexity is dependent of the implementation used to compute the eigenvalues. In our case the complexity is given as $O(knm)$, where $n$ is the size of the matrix, $k$ is the number of computed eigenvalues, and $m$ is the number of Lanczos iterations.

[4]Source and sink nodes are respectively not computed over the first and last layer.

**Neighbor Connectivity.** To extend the Local Connectivity metrics, which compute edge and node properties as a quantity w.r.t. the whole graph, we propose a set of metrics that quantify the graph connectivity throughout the nodes' degree information. The first metric computes the size of the neighborhood for each node in the graph. This is the *k-hop neighbor*, which is denoted $\mathcal{N}_k(a)$ and is the total number of nodes that are reachable in $k$ hops from the root node $a$. To have a single numerical value, we calculate $\mathcal{N}_k$ as the average $\mathcal{N}_k(a)$ over all the graph nodes. In our experiments, we decided to focus on $k = 1, 2$ to capture the connectivity both in the same layer (BGEs) and between consecutive layers (MGE), *i.e.*, ⑥ $\mathcal{N}_1$ and ⑦ $\mathcal{N}_2$, but in principle this metric can be extended to any $k$ lower than the number of layers. We then propose a metric that does not evaluate the nodes' connectivity but rather computes the different and recurrent patterns of connections. In particular, we use ⑧ $\text{Motif}_{(4)}$, *i.e.*, the sum of the number of occurrences of any *motif* of size 4 (we decided to use the size of 4 since it captures connections among consecutive layers). The motifs, which are defined as statistically significant recurrent subgraphs, have been computed using the FANMOD algorithm (Wernicke & Rasche, 2006).

**Strength Connectivity.** The previous metrics capture the node connectivity in a fixed subgraph size (*i.e.*, hops—$k$—and size of motifs). To better capture the node connectivity of the complete weighted graph structure, we introduce ⑨ the *node-strength*, defined as $\mathcal{S}(a) = \sum_{j=1}^{N} w_{a,j}$ where $w_{a,j} \in E$ represents the weight of the edge connecting nodes $a$ and $j$. We also introduce a centrality metric, ⑩ the *k-core* decomposition computed using Batagelj & Zaversnik (2003). Given a graph $G$, this is defined as its maximal subgraph $\tilde{G} = (\tilde{V}, \tilde{E})$ such that $\forall a \in \tilde{V} : \deg_{\tilde{G}(a)} \geq k$. Then, we define $k\text{-core}(a)$ as the highest order of the core ($k$) that contains $a$ (*i.e.*, the maximal degree of the subgraph that contains $a$). Both metrics are the average over all the graph nodes.

**Global Connectivity.** These metrics are computed over the complete MGE and are able to capture the overall connectivity of the graph. They (1) are more expensive computationally, but (2) can better analyze the connectivity of the networks. These topometrics are: ⑪ the number of weakly *connected components* (a.k.a. clusters), $\mathcal{C}$, and ⑫ the *average size of the clusters*, $\mathcal{C}_{avg}$, computed as $\frac{1}{|\mathcal{C}|} \sum_{i}^{|\mathcal{C}|} |c_i|$, where $c_i$ is the $i$-th component of the networks. Moreover, we identify two Global Connectivity metrics based on edges and nodes, namely ⑬ the number of *cut-edges*, and ⑭ the number *cut-nodes*, that are respectively the number of edges and nodes required to disconnect connected components. The latter, as well as *connected components*, have been computed using Depth-first search (DFS).

**Expansion.** In graph theory, it is standard to analyze the connectivity of a given graph based on its adjacency and Laplacian matrices (Coja-Oghlan, 2007; Tikhomirov & Youssef, 2016; Hoffman et al., 2021). Given a graph $G = (V, E)$, its weighted adjacency matrix is defined as $\mathcal{A}_G \in \mathbb{R}^{|V| \times |V|}$ where each entry $\mathcal{A}_G^{u,v}$ corresponds to edge weights $|E_{u,v}|^5$. The Laplacian Matrix, instead, is defined by taking into consideration the nodes' degrees as $\mathcal{L}_G = \mathcal{A}_G - \mathcal{D}$ where $\mathcal{D}$ is the degree matrix. We leverage these two matrices by extracting their $n$ eigenvalues, formally $\lambda(\mathcal{A}_G) = \{\mu_0, \mu_1, \dots, \mu_n\}$ s.t. $\mu_0 \geq \mu_1, \geq \dots \geq \mu_n$ and $\lambda(\mathcal{L}_G) = \{\mu_0', \mu_1', \dots, \mu_n'\}$ s.t. $\mu_0' \geq \mu_1' \geq \dots \geq \mu_n' = 0$, using Lehoucq et al. (1998), to compute: ⑮ the *Spectral Gap*[6], defined as $\gamma(\mathcal{A}_G) = \mu_0 - \hat{\mu}$, where $\hat{\mu} = \max_{\mu_i \neq \mu_0} \mu_i$, and ⑯ the *Spectral Radius*, computed as $\rho(\mathcal{L}_G) = \mu_0'$ (Preciado et al., 2013). For both metrics, higher values correspond to more connected structures.

## 4 Experiments

**Experimental Setup.** To provide insights into the topological properties of SNNs, we first generate a large, heterogeneous pool of sparse topologies. The graph size of the proposed unrolled MGE is based on the shape of the input data, so we select three datasets sharing the input dimensionality: CIFAR-10, CIFAR-100 (Krizhevsky et al., 2009) and the downscaled Tiny-ImageNet (Chrabaszcz et al., 2017), all with $32 \times 32$ images. We employ four different architectures: Conv-6 (Frankle & Carbin, 2019), Resnet-20 and Resnet-32 (He et al., 2016), and Wide-Resnet-28-2 (Zagoruyko & Komodakis, 2016). For pruning, we use a total of seven algorithms: four PaI methods (**SNIP** (Lee et al., 2019), **GraSP** (Wang et al., 2020), **Synflow** (Tanaka et al., 2020), **ProsPr** (Alizadeh et al., 2022)), and three instances of Layer-wise Random Pruning, namely **Uniform** (Zhu & Gupta, 2017), **ER** (Mocanu et al., 2017), and **ERK** (Evci et al., 2020).

---

[5]When computing this matrix, we use undirected graphs and the absolute values of the weights, as in Pal et al. (2022).

[6]A similar metric has been used in Hoang et al. (2023a), from a layer-based perspective. Here, we compute the spectral gap w.r.t. to the whole sparse architecture.

We apply the aforementioned algorithms at five sparsity values, to cover a broad spectrum, namely $s \in [0.6, 0.8, 0.9, 0.95, 0.98]$ (as in Hoang et al. (2023b)). We train each combination of ⟨pruning algorithm, dataset, architecture, sparsity⟩ for 3 runs, obtaining a pool of $1,260$ sparse architectures, which are characterized by the 16 topometrics defined above[7]. More details, as well as reproducibility experiments of all pruning algorithms in all settings (all successful), are reported in Appendices A and B.

### 4.1 Ramanujan-based metrics and network density

In this Section, we experiment with the metrics associated with *rolled* BGE for SNNs: the *difference of bound* (Pal et al., 2022), its follow-up relaxation, called *iterative mean difference of bound* (Hoang et al., 2023b), and the *iterative mean weighted spectral gap* (Hoang et al., 2023a). Since these metrics stem from the theory of Ramanujan graphs, we refer to them as "Ramanujan-based". Our goal here is to evaluate if these metrics provide more valuable insights than network density, which would motivate using the associated GEs in practical applications. Our findings suggest this is seldom the case.

**Preliminaries.** Given a $d$-regular graph $\mathcal{G}$, its *difference of bound* is formally defined as $\Delta r = 2 \times \sqrt{d-1} - \hat{\mu}(\mathcal{G})$. Here, $d$ is the graph regularity and $\hat{\mu}(\mathcal{G})$ denotes the largest magnitude among the non-trivial eigenvalues of the adjacency matrix of $\mathcal{G}$ (for irregular graphs, the regularity $d$ is commonly replaced by the estimated degree $d_{\mathrm{avg}}$ (Hoory, 2005)). This quantity gained popularity in the analysis of PaI algorithms thanks to its close tie with the Expansion properties of $\mathcal{G}$ itself. A value $\Delta r \geq 0$ ensures that the Ramanujan property is held by $\mathcal{G}$. In turn, for irregular graphs, this requires that $\hat{\mu}(\mathcal{G}) \leq 2 \times \sqrt{d_{\mathrm{avg}} - 1}$, which is unlikely in high sparsity regimes and would discard valid Expanders from the analysis. Thus, to cope with the strict upper bound requirement of $\hat{\mu}(\mathcal{G})$ for irregular graphs, Hoang et al. (2023b) further propose a relaxation: $\Delta r_{imdb} = \frac{1}{|K|} \sum_i^{|K|} (2 \times \sqrt{d_i - 1} - \hat{\mu}_i(K_i))$, where the difference of bound is averaged over $K$, *i.e.*, the set of *regular subgraphs* located in $\mathcal{G}$. The resulting metric is coined as the *iterative mean difference of bound*. However, both $\Delta r$ and $\Delta r_{imdb}$ ignore the values of the parameters of the model. To include this knowledge, Hoang et al. (2023a) follows the same relaxation principle of $\Delta r_{imdb}$, but introduces the *iterative mean weighted spectral gap*, defined as $\lambda_{imsg} = \frac{1}{|K|} \sum_i^{|K|} (\mu_0(|\mathcal{A}_{\mathcal{G}_i}|) - \hat{\mu}_i(|\mathcal{A}_{\mathcal{G}_i}|))$. In this context, $\mathcal{A}_{\mathcal{G}}$ denotes the weighted adjacency matrix of a graph.

All three metrics ($\Delta r$, $\Delta r_{imdb}$, and $\lambda_{imsg}$) exhibit good correlation with the SNN performance, thus representing valid candidates for performance ranking and prediction. However, graph connectivity plays a central role in the theory of Ramanujan graphs and, as a consequence, these quantities may also strongly correlate with network density, which can be trivially computed directly from the SNN as per layer $\frac{|W_{\mathrm{non-zero}}|}{|W|}$. Hence, we are interested in answering two crucial questions: **(Q1)** *To what extent are $\Delta r$, $\Delta r_{imdb}$, and $\lambda_{imsg}$ actually linked to network density?*; and **(Q2)** *Are $\Delta r$, $\Delta r_{imdb}$, and $\lambda_{imsg}$ more advantageous than density metrics to rank and/or predict the performance of SNNs?*

**Ramanujan-based metrics vs. network density.** Here we answer question **(Q1)**: what is the link between the Ramanujan-based metrics and network density? Since $\Delta r$, $\Delta r_{imdb}$, and $\lambda_{imsg}$ are computed separately for each BGE (each one encoding a different network layer), we compare them with the corresponding *layer-wise density* (*i.e.*, the fraction of active parameters in a layer) induced by PaI algorithms (for brevity, we refer to it as Layer-Density). To avoid biasing our observations towards only one architecture, and to account for the possible presence of residual connections, we make this comparison on Conv-6 and Resnet-32, as shown in Figure 2. We report the corresponding correlation values in Appendix D.

Both Figure 2(a) and Figure 2(b) convey the same message: all Ramanujan-based metrics follow the same trend of the Layer-Density and are, thus, strongly linked to the overall pruning percentage. In other words, while being intuitive, these metrics provide essentially the same information provided by the Layer-Density.

**Analyzing the performance of SNNs: $\Delta r$, $\Delta r_{imdb}$ and $\lambda_{imsg}$ vs. network density.** We now answer question **(Q2)**. Ramanujan-based metrics have been shown to correlate well with performance, *i.e.*, SNNs whose BGEs have larger $\Delta r$, $\Delta r_{imdb}$, and $\lambda_{imsg}$ tend to have better accuracy once fine-tuned. Consequently, given a ⟨model, dataset, sparsity⟩ triplet, this useful piece of information can be used to *rank* PaI methods. However, this largely holds also for the overall network density: the denser the network, the greater the

---

[7]Each topometric is normalized based on the number of nodes/edges present in the graph representation to prevent graph size/network density from being a confounding variable for our topological study.

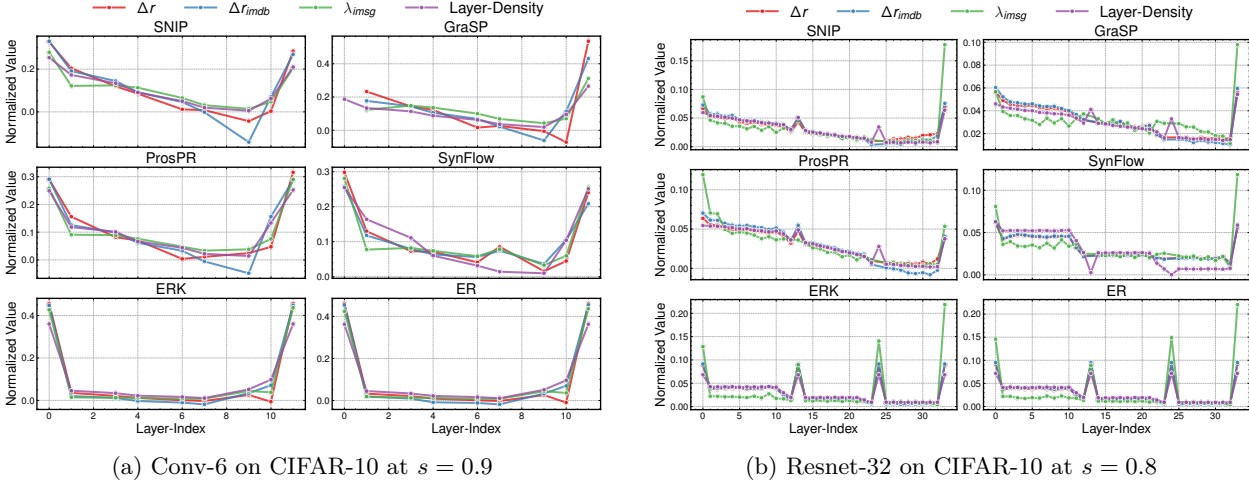

(a) Conv-6 on CIFAR-10 at $s = 0.9$  (b) Resnet-32 on CIFAR-10 at $s = 0.8$

Figure 2: Comparison between the Ramanujan-based metrics and the Layer-Density. To enable a visual comparison, we scale all values by their overall sum across layers s.t. they sum to 1.

performance. As a consequence of these observations, we aim to understand whether using $\Delta r$, $\Delta r_{imdb}$, or $\lambda_{imsg}$ gives better rankings than using the average network density across layers. The results of our analysis, shown in Table 4, provide a significant finding: *the average layer-wise sparsity is mostly comparable to, and sometimes better than, Ramanujan-based metrics as a performance ranking measure.*

While these results may suggest that SNN-derived BGEs are of limited practical interest, we argue that a key ingredient could still be missing: more informative graph metrics. Henceforth, in the next Section, we go beyond the Ramanujan-based metrics and show that large benefits to SNN analysis emerge when accounting for a larger, more informative set of graph features.

## 4.2 Accuracy drop of SNNs

In this Section, we experiment with the proposed MGE and topometrics. First, in Section 4.2.1 we show the potential of our MGE, which allows computing our proposed graph-based topological metrics over SNNs. The information extracted from these metrics is then used as a means to regress the *accuracy drop* of the SNNs with respect to their corresponding DNNs (hereafter, we always refer to the accuracy drop as $\downarrow acc = 1 - \frac{acc_s}{acc_d}$, where $acc_s$ and $acc_d$ denote the SNN and DNN accuracy, respectively). Then, we focus on understanding the relationship between the proposed topological features and the accuracy drop in Section 4.2.2. We divide all the experiments into two orthogonal scenarios.

In the **Sparsity-Fixed** scenario, we aim at understanding which topological features are statistically associated with the accuracy drop of SNNs when the computational budget (*i.e.*, the sparsity) is fixed. Importantly, we do not limit our analysis to a specific architecture, but rather we aim at general features that are important across architectural designs.

In contrast, in the **Architecture-Fixed** scenario we fix the architecture and examine how the topometrics-vs.-accuracy relationships evolve when increasing the sparsity ratio. Intuitively, we wish to determine if specific architectural designs lead to striking differences in the relevance of topological features.

### 4.2.1 Regression Analysis

We argue that the more informative the GE is, the easier it is to determine the accuracy drop of the SNNs it encodes. Hence, in the following we compare the informativeness of different GEs by regressing the accuracy drop, using standard linear least-squares regression. Given an SNN, its respective GE, and the topological features therefrom, the task entails fitting a regression model to predict the accuracy drop between the SNN and its dense counterpart.

**Baselines.** We compare our *unrolled* MGE with two encodings used as baselines, namely (1) the *rolled* BGE used in (Pal et al., 2022; Hoang et al., 2023b;a), and (2) the *rolled-channel* BGE from Prabhu et al. (2018); Vahedian et al. (2021). Then, from all types of encodings, we compute whenever possible the topometrics described in Section 3.3. It should be noted indeed that our MGE allows to extract end-to-end topological features that cannot be computed otherwise with rolled BGEs. Specifically, our encoding enables the extraction of 6 additional features w.r.t. the existing rolled BGEs: the five Local Connectivity metrics, and the 2-hop Neighbor Connectivity $(\mathcal{N}_2)$[8].

**Experimental setup.** Instead of focusing on a unique regression model, we average the regression performance over 8 different models, to provide more stable and reliable results. All of these 8 regression methods are listed in Appendix A.6. For each regressor, we employ $k$-fold cross-validation ($k = 5$), and aggregate the results over 100 runs. The numerical values used as input of the regression analysis (*i.e.* the different topometrics) have been normalized separately in $[0, 1]$ using min-max scaling. Regression performance is given in terms of the Adjusted-$R^2$ coefficient, which has been shown to be the most informative measure for discovering associations between input features and predicted variables (Chicco et al., 2021), and the Mean Absolute Error (MAE), which is a standard performance metric to compute the closeness between correct and predicted results. Both the train and test splits, obtained with $k$-fold cross-validation, contain a mixture of accuracy drop values measured on CIFAR-10, CIFAR-100, and Tiny-ImageNet. To provide generalizable conclusions and avoid inducing dataset-dependent behaviors, we do not separate the data points by dataset but use a mixture of datasets in all train and test splits.

**Results.** The results are reported in Table 3, for both the **Sparsity-Fixed** and the **Architecture-Fixed** scenarios. Our unrolled MGE outperforms the rolled BGEs for all cases in both scenarios, suggesting that our MGE is indeed more informative than existing rolled BGEs. Notably, in the **Sparsity-Fixed** scenario, the unrolled MGE exhibits stable results across sparsity ratios. This hints that the knowledge encoded in our MGE is richer than sole network density. In contrast, the information encoded in rolled BGEs is much more sensitive to changes in the overall sparsity. Finally, the results show that the gap among all methods is smaller in the **Architecture-Fixed** case, which suggests that varying the architectural design is more relevant than varying the computational budget, regardless of the encoding. Note that this is also suggested by the overall better results in the **Architecture-Fixed** case.

Table 3: $\downarrow acc$ regression analysis for both the **Sparsity-Fixed** and the **Architecture-Fixed** scenarios. We avoid reporting the MAE standard deviation since, with 3 digits approximation, it always equals 0.

| Metric | Encoding | Sparsity-Fixed | | | | | Architecture-Fixed | | | |
|---|---|---|---|---|---|---|---|---|---|---|
| | | 0.6 | 0.8 | 0.9 | 0.95 | 0.98 | Conv-6 | Resnet-20 | Resnet-32 | Wide-Resnet-28-2 |
| **Adjusted-$R^2$** (↑) | **Rolled BGE** | $.02 \pm .01$ | $.11 \pm .02$ | $.19 \pm .02$ | $.19 \pm .02$ | $.28 \pm .01$ | $.65 \pm .01$ | $.75 \pm .00$ | $.70 \pm .01$ | $.70 \pm .01$ |
| | **Rolled-channel BGE** | $.10 \pm .02$ | $.16 \pm .03$ | $.25 \pm .03$ | $.29 \pm .02$ | $.47 \pm .01$ | $.66 \pm .03$ | $.54 \pm .01$ | $.60 \pm .01$ | $.72 \pm .01$ |
| | **Unrolled MGE (ours)** | $\mathbf{.54 \pm .01}$ | $\mathbf{.58 \pm .03}$ | $\mathbf{.59 \pm .01}$ | $\mathbf{.50 \pm .01}$ | $\mathbf{.59 \pm .01}$ | $\mathbf{.72 \pm .01}$ | $\mathbf{.78 \pm .01}$ | $\mathbf{.72 \pm .01}$ | $\mathbf{.83 \pm .01}$ |
| **MAE** (↓) | **Rolled BGE** | .020 | .047 | .079 | .122 | .165 | .043 | .082 | .091 | .045 |
| | **Rolled-channel BGE** | .020 | .047 | .076 | .111 | .144 | .040 | .119 | .113 | .045 |
| | **Unrolled MGE (ours)** | **.014** | **.032** | **.056** | **.091** | **.113** | **.035** | **.080** | **.089** | **.035** |

### 4.2.2 Feature Importance

In the previous Section, we empirically verified that our unrolled MGE extracts more meaningful topological information about SNNs compared to existing rolled BGEs. However, the relationship between the topometrics therefrom and performance remains unclear. Thus, we assess here the *importance* of each topometric, examining both the **Sparsity-Fixed** and the **Architecture-Fixed** scenarios. Towards this goal, we compute the Pearson correlation coefficient (commonly denoted by $r$) between each topometric and the output of each regressor employed in the previous Section, then display the average outcome in Figures 3 and 4.

**Remark.** Note that all regressors have been fitted to predict the accuracy drop. As a consequence, when increasing a feature with a positive correlation coefficient ($r > 0$), then $\downarrow acc$ would likely increase as well (*i.e.*, the SNN would perform worse), and vice versa for features with $r < 0$.

---

[8]Moreover, we should remark that the number of motifs (Motif) is bound at 3 for the rolled BGEs (since these encodings are based on directed bipartite graphs) while is set at 4 for our MGE (as explained in Section 3).

**Sparsity-Fixed.** Figure 3 shows the feature importance (*i.e.*, the correlation coefficients), across increasing sparsity ratios. Interestingly, the importance of the Strength Connectivity and the Neighbor Connectivity topometrics decreases when increasing the sparsity. On the other hand, the Local Connectivity and the Global Connectivity connectivity metrics follow an inverse trend. An intuition for this behavior is that at lower sparsity ratios (*i.e.*, 0.6 and 0.8) the MGE is still strongly connected, thus its connectivity is better captured by nodes' degree information (*i.e.* Strength and Neighbor Connectivity metrics). Conversely, in high sparsity regimes, the network starts to be more disconnected, and here both Local and Global Connectivity have a primary role in explaining $\downarrow acc$. Importantly, these observations on Local Connectivity metrics represent a successful extension to Vysogorets & Kempe (2023); Pham et al. (2023), which introduce "effective" nodes and paths. It is, in fact, easy to see the similarities between sink, source, and "effective" nodes, as well as those between the removable in/out connections and "effective" paths. Interestingly, the correlation of the Expansion topometrics roughly remains low, without a clear pattern across sparsity ratios.

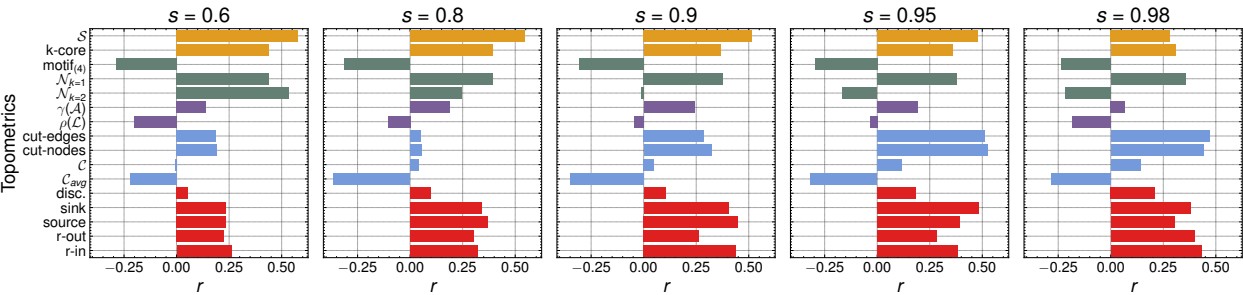

Figure 3: Pearson correlation coefficients between $\downarrow acc$ and each topometric (**Sparsity-Fixed** scenario).

**Architecture-Fixed.** This scenario is analyzed in Figure 4, from which we can make a key observation: different architectures lead to striking differences in feature importance. Hence, informative GEs, from which diverse topometrics can be extracted, are essential for the analysis. For instance, in *every* Resnet extension (*i.e.*, 20 and 32) the Local Connectivity metrics correlate strongly with the accuracy drop. In wider and more over-parameterized architectures (*i.e.*, Conv-6 and Wide-Resnet-28-2), the importance of the Strength Connectivity and Neighbor Connectivity topometrics becomes prevalent. In contrast to the **Sparsity-Fixed** scenario, Expansion topometrics generally present a strong, negative correlation with the accuracy drop. Since we do not only consider the Spectral Gap, but further examine the Spectral Radius, and compute both topometrics over the whole MGE, these findings successfully extend Hoang et al. (2023a).

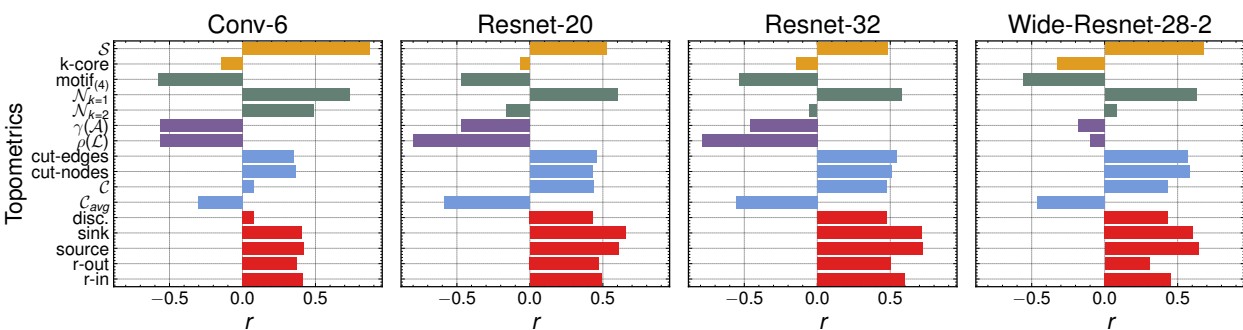

Figure 4: Pearson correlation coefficients between $\downarrow acc$ and each topometric (**Architecture-Fixed** scenario).

### 4.3 Ranking PaI algorithms via topometric mixture

In our previous analysis, we quantified the importance of each topometric for predicting the accuracy drop of SNNs. We now use feature importance for a practical aim: *ranking* PaI algorithms. Ranking entails *sorting* a set of pruning algorithms $\mathcal{P}$ according to the expected performance of their SNNs, once fine-tuned. In light of the No-Free-Lunch Theorem for pruning (Frankle et al., 2021), given a task, the optimal algorithm may save

vast amounts of computational resources. If, instead, a suboptimal PaI method is chosen, low performance is likely, and one may need to train new SNNs from scratch.

Given a ⟨model, dataset, sparsity⟩ triplet, denoted by $\langle \mathcal{M}, \mathcal{D}, s \rangle$, previous works (such as Pal et al. (2022); Hoang et al. (2023b)) rank PaI methods according to a carefully chosen topometric $x$, relying on its correlation with post-fine-tuning performance. However, this approach would discard a key finding from Section 4.2, namely that the importance of different topometrics varies with models and sparsity ratios. To close this gap, we propose to rank PaI methods with a **topometric mixture**. In detail, we propose to perform the ranking according to a *weighted sum* of topometrics, with weights embodying proxies for topometric importance in the target scenario. Specifically, we merge the importance of each topometric in the orthogonal **Sparsity-Fixed** and **Architecture-Fixed** scenarios, which allows accounting for estimates of topometric importance given both a computational budget and a network design. More formally, let $\mathbf{x}^p \in \mathbb{R}^n$ denote the $n = 16$ topometrics extracted from our MGE after encoding the SNN obtained with a pruning algorithm $p \in \mathcal{P}$, packed into a single vector. Additionally, let $\mathbf{w}^{\mathcal{M}} \in \mathbb{R}^n$ and $\mathbf{w}^s \in \mathbb{R}^n$ be the estimated **Architecture-Fixed** and **Sparsity-Fixed** feature importance vectors. We then define the overall *ranking coefficient* of $p$ as:

$$r^p = \sum_{i=0}^{n} \mathbf{x}_i^p \cdot \left( \frac{\mathbf{w}_i^{\mathcal{M}} + \mathbf{w}_i^s}{2} \right). \tag{5}$$

Since $\mathbf{w}^{\mathcal{M}}$ and $\mathbf{w}^s$ are functions of the accuracy drop, we use the proposed ranking coefficient to sort PaI algorithms in *ascending* order. Hence, for any $p_1, p_2 \in \mathcal{P}$, algorithm $p_1$ is preferred over $p_2$ iff $r^{p_1} < r^{p_2}$.

**Baselines.** We compare our ranking method to the following strategies: (1) the *difference of bound* ($\Delta_r$, Pal et al. (2022)), (2) the *iterative mean difference of bound* ($\Delta r_{imdb}$, Hoang et al. (2023b)), and (3) the *iterative mean weighted spectral gap* ($\lambda_{imsg}$) from Hoang et al. (2023a). In light of the experimental results of Section 4.1, we additionally compare our ranking strategy to (4) the *average layer-wise density*. Note that we extract $\Delta r$, $\Delta r_{imdb}$, and $\lambda_{imsg}$ from the GEs used in the respective papers.

**Experimental Setup.** For this analysis, we employ the four PaI algorithms listed at the beginning of Section 4. To measure the goodness of a ranking, we use the Rank Biased Overlap (RBO) coefficient from Webber et al. (2010), which is computed as $RBO(a, b, \alpha) = (1 - \alpha) \sum_{d=1}^{\infty} \alpha^{d-1} \frac{|a_{:d} \cap b_{:d}|}{d}$, *i.e.*, given any two ranked lists, this coefficient computes their similarity w.r.t. their length ($d$) and the order of the items. Here, $\alpha \in (0, 1)$ is a real-valued hyperparameter weighting the contribution of the top-ranked elements. To avoid biasing the results with an explicit choice for $\alpha$, we average over evenly spaced values, *i.e.*, over $\alpha \in [0.25, 0.50, 0.75]$. Following the same rationale of Section 4.2.1, we further average over CIFAR-10, CIFAR-100, and Tiny-ImageNet.

**Results.** The ranking provided in Table 4 shows that using the proposed topometric mixture outperforms all other ranking strategies on 3 out of 4 architectures (see the **Avg.** column). The only case where our proposed method struggles to reach the same performance as the competitors is Conv-6. We hypothesize that the very shallow nature of this architecture, comprised of 6 layers only, decreases the significance of end-to-end topometrics, and, in general, of GEs. In fact, the trivial average of the layer-wise density is the best performer with this architecture. When ranking deeper models, however, our MGE and the proposed topometric mixture far surpass this trivial quantity, while also largely surpassing the other ranking strategies. Finally, Table 4 lets us conclude the analysis from Section 4.1: Ramanujan-based metrics (*i.e.*, $\Delta r$, $\Delta r_{imdb}$, $\lambda_{imsg}$) are seldom more informative than layer-wise density, the only exception being the case of $s = 0.98$.

## 5 Conclusions, Limitations, and Future Directions

In conclusion, in this paper, we first identified and empirically verified the limitations of the existing layer-based GEs, as well as of the related Ramanujan-based metrics. We tackled these drawbacks with a novel MGE, modeling the SNNs in an end-to-end and input-aware manner, along with a new set of topological features for the graph-based analysis of SNNs. We extensively studied the correlation between the proposed features and the accuracy drop of SNNs and showed that an effective strategy to rank PaI methods can be derived based on feature importance, paving the way toward the adoption of SNN-based GEs in practical use cases. We showed how no topometric alone can accurately predict the performance of an SNN, but a broader

Table 4: Mean RBO of the proposed topometric mixture (**Ours**) vs. existing ranking strategies. In total, each strategy must rank seven pruning algorithms, for all ⟨model, sparsity⟩ combinations. We provide average, per-model results in the **Avg.** column.

| Model | Strategy | Sparsity | | | | | Avg. |
|---|---|---|---|---|---|---|---|
| | | 0.6 | 0.8 | 0.9 | 0.95 | 0.98 | |
| **Conv-6** | Layer-Density | **.56 ± .26** | **.58 ± .34** | **.88 ± .05** | **.88 ± .06** | .68 ± .30 | **.72 ± .20** |
| | $\Delta r$ | .44 ± .29 | .44 ± .29 | .79 ± .02 | .83 ± .03 | .84 ± .01 | .67 ± .13 |
| | $\Delta r_{imdb}$ | .49 ± .26 | .58 ± .34 | .82 ± .05 | .83 ± .04 | .80 ± .02 | .70 ± .14 |
| | $\lambda_{imsg}$ | .17 ± .00 | .20 ± .05 | .58 ± .35 | .82 ± .04 | **.84 ± .00** | .52 ± .09 |
| | **Ours** | .51 ± .29 | .51 ± .28 | .36 ± .01 | .39 ± .01 | .80 ± .01 | .51 ± .12 |
| **Resnet-20** | Layer-Density | **.95 ± .01** | .31 ± .14 | .18 ± .06 | .15 ± .04 | .18 ± .01 | .35 ± .05 |
| | $\Delta r$ | .91 ± .06 | .23 ± .14 | .15 ± .03 | .14 ± .01 | .52 ± .37 | .39 ± .12 |
| | $\Delta r_{imdb}$ | .91 ± .06 | .25 ± .12 | .15 ± .03 | .14 ± .01 | .52 ± .37 | .39 ± .12 |
| | $\lambda_{imsg}$ | .91 ± .06 | .56 ± .25 | .18 ± .09 | .14 ± .01 | .48 ± .41 | .45 ± .16 |
| | **Ours** | .57 ± .28 | **.92 ± .03** | **.44 ± .07** | **.62 ± .32** | **.92 ± .07** | **.69 ± .15** |
| **Resnet-32** | Layer-Density | .58 ± .33 | .17 ± .04 | .18 ± .10 | .15 ± .04 | .18 ± .01 | .25 ± .10 |
| | $\Delta r$ | .56 ± .36 | .16 ± .04 | .14 ± .04 | .15 ± .04 | .46 ± .40 | .29 ± .18 |
| | $\Delta r_{imdb}$ | .56 ± .36 | .16 ± .04 | .14 ± .04 | .15 ± .04 | .50 ± .37 | .30 ± .17 |
| | $\lambda_{imsg}$ | .72 ± .44 | **.77 ± .25** | .14 ± .04 | .15 ± .04 | .68 ± .36 | .49 ± .23 |
| | **Ours** | **.96 ± .03** | .60 ± .29 | **.53 ± .39** | **.61 ± .33** | **.79 ± .34** | **.70 ± .28** |
| **Wide-Resnet-28-2** | Layer-Density | .31 ± .06 | .31 ± .07 | **.48 ± .33** | .40 ± .38 | .25 ± .11 | .35 ± .19 |
| | $\Delta r$ | .28 ± .08 | .28 ± .08 | .27 ± .07 | .20 ± .12 | .21 ± .11 | .25 ± .09 |
| | $\Delta r_{imdb}$ | .28 ± .08 | .29 ± .08 | .24 ± .09 | .19 ± .09 | .21 ± .11 | .24 ± .09 |
| | $\lambda_{imsg}$ | .28 ± .08 | **.69 ± .31** | .30 ± .06 | .14 ± .01 | .21 ± .11 | .32 ± .11 |
| | **Ours** | **.35 ± .10** | .59 ± .35 | .32 ± .07 | **.52 ± .34** | **.38 ± .15** | **.43 ± .20** |

graph-based viewpoint, based on multiple topometrics, turns out to be the key to success for predicting, and then ranking, PaI performance across different sparsity ratios and models.

Despite the many advantages, a limitation of our proposed GE is the time and space complexity for the bipartite encoding generation: for each layer of the MGE (*i.e.*, for each BGE) the time complexity is $\mathcal{O}(C_{in} \times C_{out} \times step)$, while the space complexity is $\mathcal{O}(L + R + E)$, where $step = (\frac{I_d - d_{ker}}{S} + 1)^2$ and $|E| = C_{in} \times |W \setminus \{0\}| \times step$, assuming square feature maps and kernels. Hence, when computing the BGEs concatenation to obtain the end-to-end MGE, the complexity is multiplied by the number of layers, plus any additional residual connection. Additionally, in this work we focused on the established experimental setups for the graph-based analysis of SNNs, thus examining PaI methods and CNNs, as in Pal et al. (2022); Hoang et al. (2023b). Nevertheless, analyzing different models, such as Transformers, and families of pruning algorithms will be needed to further enrich the field. Furthermore, the empirical evidence found in our observational study could provide elements for theoretical studies about SNNs, as done for instance in Malach et al. (2020); Gadhikar et al. (2023). Finally, while devising pruning algorithms from graph-based SNN representations (*e.g.*, by optimizing certain topometrics according to a known sparsity ratio and model design) was out of the scope of this work, this may as well be a relevant research direction.

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

# A    Experimental Setup

## A.1    Datasets

**CIFAR-10/100** Krizhevsky et al. (2009). These two datasets are composed of $60,000$ color images each. These images are divided, respectively for the two datasets, into 10 and 100 classes. Each image has size of $32 \times 32$ pixels. On both datasets, $50,000$ images are used for training, while the remaining $10,000$ are used for testing purposes.

**Tiny-ImageNet** Chrabaszcz et al. (2017). The Tiny-ImageNet dataset consists of $100,000$ color images of size $64 \times 64$ pixels. These images are divided into 200 classes, with each class containing 500 images. The dataset is further split into three sets: a training set, a validation set, and a test set. Each class has 500 training images, 50 validation images, and 50 test images. In our experiments, we further downsample the images to $32 \times 32$ pixels in order to use the same architectures for all datasets. The downsampling has been devised using the Box algorithm [9] with the same approach used in Chrabaszcz et al. (2017).

## A.2    Architectures

**Conv-6.** This model is a scaled-down variant of VGG Simonyan & Zisserman (2014) with six convolutional and three linear layers. Max-Pooling operation is employed every two convolutional layers. Batch norm Ioffe & Szegedy (2015) is employed after every convolutional/linear layer. This model was devised for the first time in Frankle & Carbin (2019) and used as a benchmark in many recent studies on sparse models Zhou et al. (2019); Ramanujan et al. (2019); Pal et al. (2022). The total number of learnable weights in the model is $\sim 2.3$M.

**Resnet-20/Resnet-32** This model is a modification of the classical ResNet architecture. Originally proposed in He et al. (2016), it has been designed to work with images of size $32 \times 32$ pixels. The implementation is based on a public codebase[10]. This model has been benchmarked in Frankle et al. (2021); Liu et al. (2022); Alizadeh et al. (2022); Sreenivasan et al. (2022). The total number of learnable weights is $\sim 270$K for Resnet-20 and $\sim 460$K for Resnet-32.

**Wide-Resnet-28-2.** This model is a modification of Wide-ResNet, and was originally proposed in Zagoruyko & Komodakis (2016). The implementation is based on a public codebase[11]. This model has been benchmarked in Mostafa & Wang (2019); Dettmers & Zettlemoyer (2019); Frankle et al. (2021); Sreenivasan et al. (2022). The total number of learnable weights in the model is $\sim 1.4$M.

Table 5: Architectures of the models used in our experiments.

| Layer | Conv-6 | Resnet-20 | Resnet-32 | Wide-Resnet-28-2 |
|---|---|---|---|---|
| Conv 1 | | $3 \times 3, 16$ Padding 1 Stride 1 | $3 \times 3, 16$ Padding 1 Stride 1 | $3 \times 3, 16$ Padding 1 Stride 1 |
| Layer stack1 | $\begin{bmatrix} 3 \times 3, 64 \\ 3 \times 3, 64 \end{bmatrix}$ Max-Pool | $\begin{bmatrix} 3 \times 3, 16 \\ 3 \times 3, 16 \end{bmatrix} \times 3$ | $\begin{bmatrix} 3 \times 3, 16 \\ 3 \times 3, 16 \end{bmatrix} \times 5$ | $\begin{bmatrix} 3 \times 3, 32 \\ 3 \times 3, 32 \end{bmatrix} \times 4$ |
| Layer stack2 | $\begin{bmatrix} 3 \times 3, 128 \\ 3 \times 3, 128 \end{bmatrix}$ Max-Pool | $\begin{bmatrix} 3 \times 3, 32 \\ 3 \times 3, 32 \end{bmatrix} \times 3$ | $\begin{bmatrix} 3 \times 3, 32 \\ 3 \times 3, 32 \end{bmatrix} \times 5$ | $\begin{bmatrix} 3 \times 3, 64 \\ 3 \times 3, 64 \end{bmatrix} \times 4$ |
| Layer stack3 | $\begin{bmatrix} 3 \times 3, 256 \\ 3 \times 3, 256 \end{bmatrix}$ Max-Pool | $\begin{bmatrix} 3 \times 3, 64 \\ 3 \times 3, 64 \end{bmatrix} \times 3$ Avg Pool kernel size 8 | $\begin{bmatrix} 3 \times 3, 64 \\ 3 \times 3, 64 \end{bmatrix} \times 5$ Avg Pool kernel size 8 | $\begin{bmatrix} 3 \times 3, 128 \\ 3 \times 3, 128 \end{bmatrix} \times 4$ Avg Pool kernel size 8 |
| FC | $256, 256, n_{classes}$ | $64, n_{classes}$ | $64, n_{classes}$ | $128, n_{classes}$ |

---

[9] https://patrykchrabaszcz.github.io/Imagenet32/
[10] https://github.com/akamaster/pytorch_resnet_cifar10/blob/master/resnet.py
[11] https://github.com/xternalz/WideResNet-pytorch/blob/master/wideresnet.py

### A.3 Hyperparameters

For the experiments with PaI and Layer-wise Random Pruning algorithms, as well as for the dense baselines, for CIFAR-10 and CIFAR-100 we trained the models for 160 epochs with SGD with momentum 0.9 using batch size of 128. The initial learning rate was set to 0.1 and decreased by 10 at epochs 80 and 120. Weight decay was set to $1 \times 10^{-4}$ for all models but Wide-Resnet-28-2, where it was set to $5 \times 10^{-4}$. For Tiny-ImageNet, we trained the model for 100 epochs. The initial learning rate was set to 0.1 and decreased by 10 at epochs 30, 60, and 80. Momentum and weight decay hyperparameters were set as for CIFAR-10/100. This experimental setup is based on the ones used in Wang et al. (2020); Tanaka et al. (2020); Liu et al. (2022). In all cases, the weights have been initialized using the standard Kaiming Normal He et al. (2015). Concerning the input data, random cropping $(32 \times 32, \text{padding } 4)$ and horizontal flipping have been used for data augmentation in all the experiments.

### A.4 Pruning Algorithms

Details of the tested Pruning Algorithms are listed in Table 6 for PaI algorithms and in Table 7 for Layer-wise Random ones.

Table 6: Pruning at Initialization Algorithms.

| Pruning Method | Drop | Sanity Check | Training Data | Iterative |
|---|---|---|---|---|
| SNIP Lee et al. (2019) | $\lvert\nabla_\theta L(\theta)\rvert$ | ✗ | ✓ | ✗ |
| GraSPWang et al. (2020) | $-H\nabla_\theta L(\theta)$ | ✗ | ✓ | ✗ |
| Synflow Tanaka et al. (2020) | $\frac{\partial \mathcal{R}}{\partial \theta}\theta, \mathcal{R} = 1\top(\prod_{l=1}^{L}\lvert\theta^l\rvert)1$ | ✗ | ✗ | ✓ |
| ProsPr Alizadeh et al. (2022) | $\lvert\nabla_{\theta_e} L(\theta_e)\rvert$ | ✗ | ✓ | ✓ |

Table 7: Layer-wise Random Pruning Algorithms.

| Algorithm | Layer-wise Sparsity |
|---|---|
| Uniform Zhu & Gupta (2017) | $s^l \ \ \forall \ l \in [0, N)$ |
| ER Mocanu et al. (2017) | $1 - \frac{n^{l-1}+n^l}{n^{l-1}\times n^l}$ |
| ERK Evci et al. (2020) | $1 - \frac{n^{l-1}+n^l+w^l+h^l}{n^{l-1}\times n^l\times w^l\times h^l}$ |

### A.5 Implementation

In order to replicate the results of all the pruning algorithms used in this paper, we based our code on the original PyTorch algorithm implementation whenever possible. The used sources are the following:

- SNIP `https://github.com/mil-ad/snip`

- GraSP `https://github.com/alecwangcq/GraSP`

- SynFlow `https://github.com/ganguli-lab/Synaptic-Flow`

- ProsPR `https://github.com/mil-ad/prospr`

- Uniform, ER and ERK `https://github.com/VITA-Group/Random_Pruning`

On the other hand, for the IMDB metric and the rolled graph encoding Hoang et al. (2023b) we relied on the official implementation available at `https://github.com/VITA-Group/ramanujan-on-pai`.

### A.6 Models used in regression analysis of performance drop of SNN

In the performance drop analysis carried out in Section 4.2 we tested the topometrics over 8 different regressor model namely: ① Linear Model, ② Ridge, ③ Lasso, ④ ElasticNet, ⑤ Huber, ⑥ Principal Component Regression, ⑦ Bayesian Ridge, and ⑧ Automatic Relevance Determination (ARD).

# B  Results

In this section, the results of the experiments explained in Appendix A are shown separately for each dataset, see Tables 8-10. Each combination ⟨dataset, architecture, sparsity, pruning algorithm⟩ has been evaluated over 3 runs. In the tables, the symbol "*" indicates that the sparse network is not able to outperform the random dense baseline (*i.e.*, if the probability of correctly guessing a sample class is $\frac{1}{C}$, the network after pruning has an average accuracy $\leq \frac{1}{C}$). In each table, for any combination ⟨architecture, sparsity⟩.

Table 8: Accuracy results on CIFAR-10.

| | Conv-6 | | | | | Resnet-20 | | | | | Resnet-32 | | | | | Wide-Resnet-28-2 | | | | |
|---|---|---|---|---|---|---|---|---|---|---|---|---|---|---|---|---|---|---|---|---|
| Sparsity Ratio | 0.6 | 0.8 | 0.9 | 0.95 | 0.98 | 0.6 | 0.8 | 0.9 | 0.95 | 0.98 | 0.6 | 0.8 | 0.9 | 0.95 | 0.98 | 0.6 | 0.8 | 0.9 | 0.95 | 0.98 |
| SNIP Lee et al. (2019) | 92.7±0.3 | 92.0±0.1 | 91.1±0.1 | 89.6±0.2 | 83.8±0.3 | 90.5±0.4 | 89.0±0.3 | 87.0±0.0 | 83.2±0.4 | 75.9±0.5 | 91.7±0.1 | 90.5±0.2 | 88.9±0.2 | 86.3±0.1 | 80.2±0.6 | 94.1±0.2 | 93.2±0.2 | 92.2±0.2 | 90.6±0.2 | 87.2±0.3 |
| GraSP Wang et al. (2020) | 92.2±0.1 | 91.6±0.1 | 90.7±0.1 | 89.5±0.1 | 86.0±0.3 | 90.2±0.1 | 89.1±0.3 | 87.2±0.1 | 84.4±0.0 | 77.1±0.2 | 91.3±0.3 | 90.4±0.1 | 89.1±0.4 | 86.6±0.2 | 81.8±0.0 | 94.0±0.2 | 93.0±0.1 | 91.9±0.1 | 90.5±0.1 | 87.7±0.3 |
| SynFlow Tanaka et al. (2020) | 92.7±0.1 | 92.1±0.1 | 91.3±0.3 | 89.8±0.2 | 86.8±0.2 | 90.5±0.3 | 89.3±0.1 | 86.2±0.3 | 82.9±0.2 | 76.2±0.3 | 91.6±0.3 | 90.4±0.5 | 88.7±0.2 | 85.6±0.2 | 79.7±0.5 | 93.9±0.1 | 93.1±0.1 | 91.6±0.3 | 90.4±0.2 | 86.6±0.4 |
| ProsPR Alizadeh et al. (2022) | 92.5±0.3 | 91.7±0.3 | 90.5±0.3 | 89.2±0.3 | 85.3±0.5 | 90.6±0.2 | 89.4±0.1 | 86.4±0.3 | 81.8±0.5 | 74.6±0.5 | 91.9±0.1 | 90.7±0.4 | 88.4±0.4 | 84.3±1.0 | 54.4±0.0 | 94.0±0.3 | 93.4±0.1 | 92.1±0.2 | 90.5±0.3 | 87.2±0.6 |
| Uniform Zhu & Gupta (2017) | 92.3±0.2 | 91.1±0.1 | 89.2±0.1 | 86.4±0.4 | 79.7±0.1 | 90.1±0.1 | 88.2±0.4 | 85.5±0.3 | 77.7±4.2 | 54.7±6.9 | 91.2±0.3 | 89.8±0.2 | 87.8±0.4 | 81.9±5.3 | 44.1±8.0 | 93.5±0.1 | 92.5±0.0 | 90.9±0.0 | 89.0±0.1 | 84.3±0.6 |
| ER Mocanu et al. (2017) | 91.3±0.1 | 90.2±0.1 | 88.6±0.2 | 86.5±0.1 | 82.7±0.2 | 90.5±0.2 | 89.1±0.1 | 86.9±0.3 | 83.8±0.2 | 77.0±0.3 | 91.6±0.2 | 90.8±0.1 | 88.9±0.0 | 85.5±0.2 | 81.8±0.2 | 93.8±0.1 | 93.2±0.2 | 92.1±0.1 | 90.4±0.2 | 87.0±0.0 |
| ERK Evci et al. (2020) | 91.3±0.1 | 90.2±0.2 | 88.8±0.3 | 86.7±0.1 | 82.5±0.1 | 90.8±0.3 | 89.4±0.3 | 87.1±0.1 | 83.8±0.2 | 77.4±0.2 | 91.6±0.3 | 90.7±0.2 | 88.9±0.1 | 86.4±0.3 | 81.8±0.3 | 94.1±0.1 | 93.4±0.1 | 92.2±0.3 | 90.4±0.2 | 86.9±0.4 |
| Dense (Baseline) | 93.2±0.0 | | | | | 91.7±0.1 | | | | | 92.3±0.1 | | | | | 94.4±0.1 | | | | |

Table 9: Accuracy results on CIFAR-100.

| | Conv-6 | | | | | Resnet-20 | | | | | Resnet-32 | | | | | Wide-Resnet-28-2 | | | | |
|---|---|---|---|---|---|---|---|---|---|---|---|---|---|---|---|---|---|---|---|---|
| Sparsity Ratio | 0.6 | 0.8 | 0.9 | 0.95 | 0.98 | 0.6 | 0.8 | 0.9 | 0.95 | 0.98 | 0.6 | 0.8 | 0.9 | 0.95 | 0.98 | 0.6 | 0.8 | 0.9 | 0.95 | 0.98 |
| SNIP Lee et al. (2019) | 66.9±0.5 | 65.2±0.4 | 62.8±0.7 | 56.7±1.4 | 38.2±1.8 | 64.3±0.2 | 60.6±1.1 | 52.5±1.5 | 41.9±0.9 | 24.4±0.5 | 66.3±0.5 | 61.9±2.2 | 53.9±1.7 | 46.4±2.2 | 28.9±1.3 | 72.4±0.2 | 70.4±0.2 | 67.8±0.2 | 63.3±0.2 | 51.2±1.2 |
| GraSP Wang et al. (2020) | 66.5±0.3 | 64.5±0.4 | 62.6±0.2 | 59.6±0.1 | 52.2±0.5 | 63.7±0.3 | 60.4±0.3 | 54.2±0.3 | 45.9±0.1 | 31.2±1.3 | 66.0±0.4 | 63.3±0.2 | 59.8±0.6 | 52.8±0.4 | 38.1±0.7 | 71.9±0.3 | 69.9±0.5 | 67.7±0.3 | 64.2±0.6 | 55.1±1.1 |
| SynFlow Tanaka et al. (2020) | 67.4±0.3 | 65.1±0.0 | 63.2±0.3 | 60.5±0.2 | 54.7±0.1 | 63.9±0.1 | 58.9±0.4 | 50.1±0.1 | 38.2±0.6 | 19.2±2.5 | 66.2±0.1 | 62.5±0.2 | 55.3±0.2 | 44.0±0.3 | 24.1±1.0 | 71.8±0.2 | 69.3±0.5 | 66.1±0.4 | 60.5±0.0 | 48.7±0.3 |
| ProsPR Alizadeh et al. (2022) | 67.3±0.3 | 64.8±0.6 | 62.9±0.2 | 59.6±0.1 | 52.2±0.4 | 64.1±0.1 | 60.0±0.7 | 50.9±0.9 | 31.7±3.2 | 26.5±1.1 | 66.2±0.2 | 62.9±0.5 | 54.0±3.5 | 35.2±6.3 | 4.4±2.7 | 72.4±0.5 | 70.5±0.3 | 67.9±0.3 | 61.5±0.4 | 45.1±3.2 |
| Uniform Zhu & Gupta (2017) | 66.4±0.2 | 63.6±0.4 | 59.9±0.1 | 54.8±0.6 | 43.6±1.2 | 63.1±0.3 | 59.5±0.4 | 52.2±0.8 | 39.1±3.0 | 18.2±0.9 | 65.4±0.3 | 62.3±0.8 | 56.8±0.4 | 46.1±1.0 | 23.5±1.4 | 72.0±0.2 | 69.7±0.5 | 67.2±0.1 | 62.7±0.5 | 45.6±2.0 |
| ER Mocanu et al. (2017) | 64.8±0.3 | 62.3±0.1 | 59.9±0.4 | 55.6±0.1 | 47.5±0.8 | 64.6±0.5 | 61.4±0.2 | 55.8±0.4 | 47.2±0.5 | 33.7±0.5 | 66.4±0.3 | 64.3±0.3 | 59.7±0.3 | 53.6±0.8 | 40.9±0.5 | 72.2±0.3 | 70.2±0.4 | 68.0±0.6 | 64.1±0.4 | 56.1±0.4 |
| ERK Evci et al. (2020) | 65.1±0.5 | 62.0±0.1 | 59.0±0.4 | 56.0±0.2 | 47.3±0.1 | 65.4±0.5 | 61.8±0.2 | 55.8±1.2 | 47.2±0.1 | 33.6±0.6 | 66.7±0.6 | 64.7±0.5 | 60.2±0.4 | 53.0±0.1 | 40.8±0.5 | 72.4±0.4 | 70.7±0.1 | 68.1±0.1 | 64.7±0.6 | 56.3±0.1 |
| Dense (Baseline) | 68.5±0.2 | | | | | 66.4±0.3 | | | | | 68.0±0.3 | | | | | 74.2±0.2 | | | | |

Table 10: Accuracy results on Tiny-ImageNet.

| | Conv-6 | | | | | Resnet-20 | | | | | Resnet-32 | | | | | Wide-Resnet-28-2 | | | | |
|---|---|---|---|---|---|---|---|---|---|---|---|---|---|---|---|---|---|---|---|---|
| Sparsity Ratio | 0.6 | 0.8 | 0.9 | 0.95 | 0.98 | 0.6 | 0.8 | 0.9 | 0.95 | 0.98 | 0.6 | 0.8 | 0.9 | 0.95 | 0.98 | 0.6 | 0.8 | 0.9 | 0.95 | 0.98 |
| SNIP Lee et al. (2019) | 45.9±0.2 | 43.7±0.6 | 37.3±0.2 | 29.2±1.0 | 16.6±1.5 | 41.5±0.3 | 34.2±1.5 | 27.2±0.5 | 19.8±0.3 | 10.8±0.6 | 41.9±1.9 | 34.6±0.6 | 28.4±0.6 | 20.5±1.4 | 13.6±1.1 | 48.7±0.3 | 47.8±0.5 | 44.5±0.9 | 38.3±0.8 | 27.8±0.6 |
| GraSP Wang et al. (2020) | 45.2±0.5 | 44.5±0.4 | 42.6±0.5 | 39.0±0.3 | 31.6±0.4 | 41.1±0.4 | 36.4±0.6 | 30.5±0.2 | 23.2±1.2 | 13.0±0.4 | 44.2±0.3 | 40.3±0.7 | 34.2±0.2 | 27.7±0.1 | 17.6±0.2 | 47.8±0.8 | 46.9±0.7 | 44.0±0.3 | 39.3±0.4 | 30.8±0.7 |
| SynFlow Tanaka et al. (2020) | 46.2±0.2 | 45.0±0.3 | 43.2±0.4 | 40.0±0.4 | 32.6±0.7 | 41.4±0.1 | 33.7±0.4 | 24.4±0.3 | 13.9±0.4 | 6.1±0.8 | 44.4±0.3 | 37.6±0.7 | 27.2±0.2 | 16.9±0.8 | 7.0±1.4 | 48.5±0.5 | 46.0±0.1 | 41.2±0.4 | 33.6±0.5 | 21.2±0.9 |
| ProsPR Alizadeh et al. (2022) | 46.3±0.4 | 45.7±0.9 | 43.1±0.4 | 39.3±0.3 | 31.8±0.8 | 40.4±1.1 | 34.6±0.1 | 22.0±1.0 | 11.1±1.5 | 10.8±0.0 | 44.3±0.9 | 37.5±0.4 | 24.3±2.1 | 11.5±1.7 | * | 49.3±0.3 | 47.1±0.4 | 43.8±0.3 | 34.9±0.9 | 27.5±0.1 |
| Uniform Zhu & Gupta (2017) | 45.6±0.5 | 43.8±0.3 | 40.3±0.3 | 34.9±0.4 | 25.7±0.2 | 40.4±0.1 | 35.3±0.3 | 28.5±0.3 | 20.5±0.4 | 9.3±1.4 | 44.2±0.3 | 39.0±0.6 | 32.4±0.1 | 24.1±1.0 | 11.9±2.4 | 47.7±0.9 | 46.1±0.4 | 41.7±0.8 | 36.0±0.4 | 25.3±0.8 |
| ER Mocanu et al. (2017) | 44.4±0.4 | 43.7±0.2 | 40.3±0.4 | 36.3±0.5 | 28.5±0.4 | 43.1±0.7 | 37.8±0.3 | 32.2±0.4 | 25.6±0.2 | 15.1±0.4 | 46.6±0.2 | 42.2±0.6 | 36.1±0.3 | 29.2±0.1 | 21.1±0.5 | 49.3±0.4 | 48.0±0.3 | 44.6±0.3 | 39.9±0.4 | 31.5±0.3 |
| ERK Evci et al. (2020) | 44.1±0.3 | 43.0±0.4 | 40.9±0.1 | 35.8±0.2 | 28.6±0.3 | 43.8±0.5 | 38.6±0.4 | 32.1±0.0 | 25.4±0.2 | 15.6±0.2 | 46.3±0.1 | 42.3±0.3 | 35.8±0.4 | 28.7±0.6 | 19.6±0.5 | 48.9±0.2 | 48.1±0.2 | 44.4±0.4 | 39.6±0.5 | 31.6±0.5 |
| Dense (Baseline) | 46.1±0.4 | | | | | 46.3±0.3 | | | | | 48.0±0.5 | | | | | 48.6±0.2 | | | | |

# C  Graph Encoding Pooling Layers

Suppose the network has two consecutive layers $l_i$ and $l_{i+1}$ and a pooling operation $\rho$ is employed between them. The two layers have respectively a bipartite graph representation $G_i$ and $G_{i+1}$. However, the MGE is not directly applicable since $|R_i| \neq |L_{i+1}|$. Nevertheless, we know a priori that $|\rho(R_i)| = |L_{i+1}|$. This encoding extension aims at concatenating two layers $R_i$ and $L_{i+1}$ with different dimensions. To do so, we transform $L_{i+1}$ to $L'_{i+1}$ such that $|L'_{i+1}| = |R_i|$, to allow the nodes' concatenation between consecutive layers. Then, the edge construction is straightforward: each edge in $G_{i+1}$ which links $u \in L_{i+1}$ to $v \in R_{i+1}$ is added $|\rho(R_i)|$ times, one for each node in the pooling window $p_w \subseteq R_i$, which is computed over $R_i$ (since $R_i$ has the same dimension of $L'_{i+1}$ and the two have been concatenated) such that $u \in L'_{i+1}$ is linked to $v \in R_{i+1}$. Then, based on such pooling encoding, any pair of consecutive layers $G'_i$ and $G_{i+1}$ can be linked together in the MGE.

# D  Correlation Expander-related metrics and network density

In Figure 5, we report the Pearson Correlation ($r$) between the Ramanujan-based metrics and the Layer-Density. The correlation has been computed over the metrics on all layers of the given ⟨architectures,sparsity⟩, considering as dataset CIFAR-10. It can be seen how the correlation for $\Delta r$ Pal et al. (2022) and $\Delta r_{imdb}$ Hoang et al. (2023b) is almost always above 0.9, with several peaks at $\sim 1$, while it is slightly lower for $\lambda_{imsg}$ Hoang et al. (2023a). To note that empty entries refer to SNNs that cannot be analyzed using the Ramanujan-based metrics, since in those cases the graph representation violates the Ramanujan constraints of $\min(d_L, d_r) \geq 3$ Hoang et al. (2023b) for *every* layer.

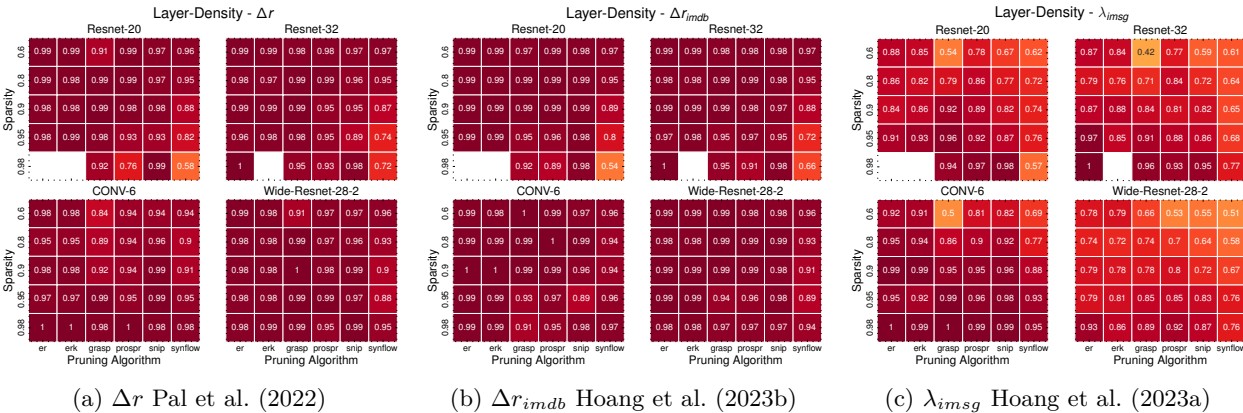

Figure 5: Pearson Correlation ($r$) between Ramanujan-based metrics and Layer-Density

# E   Rank Prediction

In this section, we include the numerical values (in terms of mean RBO) of the rank prediction devised with our topometric mixture, separately for *Pruning at Initialization* and *Layer-Wise Random Pruning* algorithms. Table 11 shows the results for the PaI case, where we can see that the numerical values follow the same pattern presented in the main text, *i.e.*, our proposed topometric mixture outdoes the other metrics by $\sim$ 30%. On the other hand, in Table 12 we report the results for the Layer-Wise Random Pruning algorithms, where no clear better approach is visible. The numerical values of the mean RBO turn out to be high for almost all metrics (ours included), which indicates almost perfect matching in the ranking prediction.

Table 11: Mean RBO of the proposed topometric mixture (**Ours**) vs. existing ranking strategies. In total, each strategy must rank four *Pruning at Initialization* algorithms (see Section 4), for all ⟨model, sparsity⟩ combinations. We provide average, per-model results in the **Avg.** column.

| Model | Encoding | Sparsity-wise | | | | | Avg. |
|---|---|---|---|---|---|---|---|
| | | 0.6 | 0.8 | 0.9 | 0.95 | 0.98 | |
| **Conv-6** | Layer-Density | **.60 $\pm$ .26** | **.62 $\pm$ .34** | **.91 $\pm$ .05** | **.90 $\pm$ .06** | .69 $\pm$ .29 | .74 $\pm$ .20 |
| | $\Delta r$ | .60 $\pm$ .26 | .62 $\pm$ .34 | .91 $\pm$ .05 | .86 $\pm$ .00 | .86 $\pm$ .00 | .77 $\pm$ .13 |
| | $\Delta r_{imdb}$ | **.60 $\pm$ .26** | **.62 $\pm$ .34** | **.91 $\pm$ .05** | .86 $\pm$ .00 | .86 $\pm$ .00 | .77 $\pm$ .13 |
| | $\lambda_{imsg}$ | **.60 $\pm$ .26** | **.62 $\pm$ .34** | **.91 $\pm$ .05** | **.90 $\pm$ .06** | .86 $\pm$ .00 | **.78 $\pm$ .14** |
| | **Ours** | .54 $\pm$ .28 | .53 $\pm$ .29 | .37 $\pm$ .02 | .34 $\pm$ .07 | **.93 $\pm$ .06** | .54 $\pm$ .14 |
| **Resnet-20** | Layer-Density | .62 $\pm$ .34 | .53 $\pm$ .29 | .36 $\pm$ .00 | .36 $\pm$ .00 | .35 $\pm$ .08 | .44 $\pm$ .14 |
| | $\Delta r$ | .42 $\pm$ .07 | .36 $\pm$ .10 | .26 $\pm$ .00 | .26 $\pm$ .00 | .33 $\pm$ .06 | .33 $\pm$ .05 |
| | $\Delta r_{imdb}$ | .42 $\pm$ .07 | .36 $\pm$ .10 | .26 $\pm$ .00 | .26 $\pm$ .00 | .33 $\pm$ .06 | .33 $\pm$ .05 |
| | $\lambda_{imsg}$ | .42 $\pm$ .07 | .36 $\pm$ .10 | .26 $\pm$ .00 | .29 $\pm$ .06 | .33 $\pm$ .06 | .33 $\pm$ .06 |
| | **Ours** | **.90 $\pm$ .06** | **.54 $\pm$ .31** | **.82 $\pm$ .28** | **.93 $\pm$ .06** | **.53 $\pm$ .29** | **.74 $\pm$ .20** |
| **Resnet-32** | Layer-Density | .55 $\pm$ .30 | **.50 $\pm$ .43** | .33 $\pm$ .06 | .36 $\pm$ .00 | .36 $\pm$ .00 | .42 $\pm$ .16 |
| | $\Delta r$ | **.72 $\pm$ .28** | **.54 $\pm$ .31** | .29 $\pm$ .06 | .26 $\pm$ .00 | .26 $\pm$ .00 | .41 $\pm$ .13 |
| | $\Delta r_{imdb}$ | **.72 $\pm$ .28** | .34 $\pm$ .07 | .29 $\pm$ .06 | .26 $\pm$ .00 | .26 $\pm$ .00 | .37 $\pm$ .08 |
| | $\lambda_{imsg}$ | .59 $\pm$ .33 | **.50 $\pm$ .34** | .29 $\pm$ .06 | .36 $\pm$ .00 | .36 $\pm$ .00 | .42 $\pm$ .15 |
| | **Ours** | .59 $\pm$ .36 | .33 $\pm$ .06 | **.93 $\pm$ .06** | **.97 $\pm$ .0** | **.73 $\pm$ .32** | **.71 $\pm$ .16** |
| **Wide-Resnet-28-2** | Layer-Density | .61 $\pm$ .22 | .44 $\pm$ .04 | **.76 $\pm$ .35** | .53 $\pm$ .29 | .38 $\pm$ .02 | .54 $\pm$ .18 |
| | $\Delta r$ | .37 $\pm$ .02 | .37 $\pm$ .02 | .36 $\pm$ .00 | .29 $\pm$ .06 | .29 $\pm$ .06 | .34 $\pm$ .03 |
| | $\Delta r_{imdb}$ | .37 $\pm$ .02 | .37 $\pm$ .02 | .33 $\pm$ .06 | .29 $\pm$ .06 | .29 $\pm$ .06 | .33 $\pm$ .04 |
| | $\lambda_{imsg}$ | .37 $\pm$ .02 | .37 $\pm$ .02 | .36 $\pm$ .00 | .29 $\pm$ .06 | .33 $\pm$ .06 | .34 $\pm$ .03 |
| | **Ours** | **.63 $\pm$ .29** | **.82 $\pm$ .31** | .56 $\pm$ .29 | **.73 $\pm$ .29** | **.94 $\pm$ .07** | **.74 $\pm$ .25** |

Table 12: Mean RBO of the proposed topometric mixture (**Ours**) vs. existing ranking strategies. In total, each strategy must rank three *Layer-Wise Random Pruning* algorithms (see Section 4), for all ⟨model, sparsity⟩ combinations. We provide average, per-model results in the **Avg.** column.

| Model | Encoding | Sparsity-wise | | | | | Avg. |
|---|---|---|---|---|---|---|---|
| | | 0.6 | 0.8 | 0.9 | 0.95 | 0.98 | |
| **Conv-6** | Layer-Density | .04 ± .00 | .40 ± .00 | **.56 ± .29** | **1.0 ± .00** | **.67 ± .29** | .61 ± .12 |
| | $\Delta r$ | .40 ± .00 | .40 ± .00 | .56 ± .29 | 1.0 ± .00 | .67 ± .29 | .61 ± .12 |
| | $\Delta_{imdb}$ | **.97 ± .06** | **.97 ± .06** | .43 ± .06 | **1.0 ± .00** | **.67 ± .29** | **.81 ± .09** |
| | $\lambda_{imsg}$ | .40 ± .00 | .40 ± .00 | **.56 ± .29** | .83 ± .29 | .63 ± .32 | .56 ± .18 |
| | **Ours** | **.97 ± .06** | **.97 ± .06** | .43 ± .06 | .83 ± .29 | **.67 ± .29** | .77 ± .15 |
| **Resnet-20** | Layer-Density | **1.0 ± .00** | **1.0 ± .00** | .83 ± .29 | **.67 ± .29** | **1.0 ± .00** | **.90 ± .12** |
| | $\Delta r$ | **1.0 ± .00** | **1.0 ± .00** | .83 ± .29 | .50 ± .00 | .83 ± .29 | .83 ± .12 |
| | $\Delta_{imdb}$ | **1.0 ± .00** | **1.0 ± .00** | .83 ± .29 | **.67 ± .29** | .83 ± .29 | .87 ± .17 |
| | $\lambda_{imsg}$ | **1.0 ± .00** | .67 ± .29 | **1.0 ± .0** | **.67 ± .29** | .67 ± .29 | .80 ± .17 |
| | **Ours** | .83 ± .29 | **1.0 ± .00** | .67 ± .29 | **.67 ± .29** | **1.0 ± .00** | .83 ± .17 |
| **Resnet-32** | Layer-Density | .83 ± .29 | **.83 ± .29** | **1.0 ± .00** | **1.0 ± .00** | .83 ± .29 | .90 ± .17 |
| | $\Delta r$ | .83 ± .29 | **.83 ± .29** | **1.0 ± .00** | .83 ± .29 | **1.0 ± .00** | .90 ± .17 |
| | $\Delta_{imdb}$ | .83 ± .29 | **.83 ± .29** | **1.0 ± .00** | .83 ± .29 | **1.0 ± .00** | **.90 ± .17** |
| | $\lambda_{imsg}$ | **1.0 ± .00** | **.83 ± .29** | **1.0 ± .00** | .83 ± .29 | **1.0 ± .00** | .93 ± .12 |
| | **Ours** | **1.0 ± .00** | .67 ± .29 | .67 ± .29 | .83 ± .29 | .83 ± .29 | .80 ± .23 |
| **Wide-Resnet-28-2** | Layer-Density | **.83 ± .29** | **1.0 ± .00** | **.83 ± .29** | **1.0 ± .00** | .50 ± .0 | .83 ± .12 |
| | $\Delta r$ | **.83 ± .29** | **1.0 ± .00** | **.83 ± .29** | .83 ± .29 | **.83 ± .29** | .86 ± .23 |
| | $\Delta_{imdb}$ | **.83 ± .29** | **1.0 ± .00** | **.83 ± .29** | **1.0 ± .00** | **.83 ± .29** | **.90 ± .17** |
| | $\lambda_{imsg}$ | **.83 ± .29** | **1.0 ± .00** | **.83 ± .29** | .83 ± .29 | .67 ± .29 | .83 ± .23 |
| | **Ours** | **.83 ± .29** | **1.0 ± .00** | .67 ± .29 | .83 ± .29 | .67 ± .29 | .80 ± .23 |

