# OpenReview forum: "Understanding Sparse Neural Networks from their Topology via Multipartite Graph Representations"
_TMLR — Accepted by TMLR_

### Review · Reviewer_nJmv · 2024-02-09

**Summary Of Contributions:**

This paper focuses on the relating the performance of pruning at initialization schemes, with the metrics developed in graph theory. The authors discovered that the existing Ramanujan-based metrics (obtaining which requires high computational complexity) often do not provide more information than the readily-available layer-wise density metric. Motivated by this finding, the authors consider measuring the metrics for unrolled input-aware Multipartite Graph Encoding (MGE) setup, which is empirically shown to have high correlation with the accuracy drop of sparse neural networks (SNNs).

**Audience:**

Yes

**Claims And Evidence:**

Yes

**Requested Changes:**

I guess the theoretical analysis is bit hard to cover, but giving some insight on the experimental results is required. Moreover, the clarification of the results and conclusion is needed.

**Strengths And Weaknesses:**

Strength
- This paper focuses on the interesting research problem of how the topological metrics of sparse neural networks (obtained by prune-at-initialization methods) are related with the performance of such pruned network.
- The idea of using unrolled input-aware MGEs makes sense.

Weakness
- The pearson correlation in Fig.3 and 4 is positive for several metrics, but we also have some metrics with negative correlation. Do we have some guideline on which metric we should rely on?
- Although this paper has a through experimental results, it is not based on some theoretical analysis.
- The claim of this paper is bit unclear. Are the authors claiming that "Regarding SNNs obtained by Pruning at Initialization methods, metric XXX is a good indicator of the performance of the SNN"? What is XXX in this claim?

---

> ### Author Response · Authors · 2024-03-16
>
> We thank the reviewer for the valuable feedback. Below are our comments about the Weaknesses section. We also updated the manuscript (the parts related to this reviewer’s comments are in orange) to clarify the following points.
>
> **The pearson correlation in Fig.3 and 4 is positive for several metrics, but we also have some metrics with negative correlation. Do we have some guideline on which metric we should rely on?**
>
> The Pearson correlation coefficient shown in Figures 3 and 4, as highlighted by the reviewer, could be either positive or negative. Since the regressor has been fitted to predict the performance drop for any topometric x, a negative correlation coefficient means that the accuracy drop is expected to decrease when increasing x. As per the remark in Sec. 4.2, this makes x an important feature to maximize, given that one would ultimately aim at decreasing the accuracy drop. The inverse holds for positive correlations. Hence, we decided on purpose to also show the sign of the correlation, rather than just its absolute value, since the sign turns out to be the key to the success of the proposed ranker based on topometric mixture. However, in the general approach, we advise relying on the metric with the higher correlation amplitude, regardless of the sign.
>
> **Although this paper has a through experimental results, it is not based on some theoretical analysis.**
>
> We are aware of emerging works providing theoretical insights about SNNs, such as [1,3]. We acknowledged their merits and added the proper references. In the Limitations Section, we will explicitly say that the focus of this work is mostly empirical. Nevertheless, the theory of SNNs is slowly emerging, and our experiments yield new hypotheses that are natural candidates for future theoretical validation within the field.
>
> **Regarding SNNs obtained by Pruning at Initialization methods, metric XXX is a good indicator of the performance of the SNN"?**
>
> The claim of this paper is not that “metric XXX is a good indicator of the performance of the SNN”, it is rather the opposite: it is almost impossible to accurately predict the performance of SNN using a single topometric (XXX). To this aim, we claim that analyzing SNNs from a broader viewpoint, i.e., using several topometrics, is much more effective.
> Understanding why different pruning algorithms perform differently w.r.t. the models, dataset, and sparsity levels has been proven to be highly challenging in several works. In [2], the no-free-lunch theorem of pruning has been introduced. In our work, we tried to complement the discovery of [2],  emphasizing the relationship between various topometrics and sparse model performance, unveiling that severe performance drops co-occur with changes in different topometrics according to the scenario. The idea of regressing both on sparsity-fixed and model-fixes settings, and then averaging the importance of different topometrics, came from the intuition that each sparsity ratio and model is associated with different important topometrics to preserve.
>
> **References**
>
> [1] Malach, Eran, et al. "Proving the lottery ticket hypothesis: Pruning is all you need." International Conference on Machine Learning. PMLR, 2020.
>
> [2] Frankle, Jonathan, et al. "Pruning Neural Networks at Initialization: Why Are We Missing the Mark?." International Conference on Learning Representations. 2020.
>
> [3] Gadhikar, Advait Harshal, Sohom Mukherjee, and Rebekka Burkholz. "Why random pruning is all we need to start sparse." International Conference on Machine Learning. PMLR, 2023.

---

### Review · Reviewer_deP5 · 2024-02-24

**Summary Of Contributions:**

The paper proposes to use a set of graph topological metrics calculated from a mutlipartie graph encodening (MGE) of a neural network to analyse and predict performance of pruned-at-initialization (PaI) sparse neural networks (SNNs). The MGE itself is also new and is build from an unrolled (input-aware) bi-partie graph encodings. The MGE construction is covered for MLP and CNN architectures including residual connections and pooling operations, architectures such as transformers remain for future work. The newly proposed topometrics are analysed in a set of experiments and compared to the previously proposed Ramanujan-based metrics and simple layer-density concluding that Ramajuan-based metrics are rarely more informative than the simple layer-density and that the proposed topometrics can provide more valuable information to understand the pruning effects.

**Audience:**

Yes

**Broader Impact Concerns:**

No concerns

**Claims And Evidence:**

Yes

**Requested Changes:**

Please see weaknesses paragraph before: complement topometrics description and provide link to code-base.

Minor change requests:
- What happens in your BGE and MGE if padding is not zero? Would these padding nodes be included in the graphs? Please clarify.
- section 3.2, second paragraph, second line, definition of $\hat{E}$ should it read $i \neq j$ or rather $i < j$?
- section 3.2, second paragraph, pooling operation graph - you refer to Appendix C. Please give at least a brief description here to help the intuition without the need to read the appendix.
- section 3.2, second paragraph, residual connections - how are these included in the graph? This is probably trivial but a diagram would help.
- r-out and r-in - are this no. (as in counts) or rather ratios? Please correct if these are ratios.
- source nodes - Do I understand correctly that these can only exist in the 1st layer? Or can they appear in some later layers due to the sparsifying operations? Please comment in the text to help the reader.
- neihbourhood connectivity - are the k-hop neihborhoods directional?
- Expansion - formal definitions of eigenvalues $\lambda(\mathcal{L}_G) = \{ \ldots \}$ has a typo in the last element of the list. Should be $n$ and not $0$.
- Spectral gap - is this correct? Isn't the $\hat{\mu}$ simply the 2nd biggest eigenvalue and hence $\mu_1$?
- section 4.1 - network density is introduced as a term which is, however, never explained. I could only understand what it is when coming to page 8 which says under Ramanujan-based metrics paragraph that it is the fraction of active parameters.
- section 4.2.1 - I believe this is a standard linear ordinary-least squares regression model. Please state it clearly to avoid confusion.
- Page 11 last paragraph - please spell out RBO before using the abbreviation.

**Strengths And Weaknesses:**

The paper is very well written and easy to follow. It explains the necessary basics to understand the setting and context at an appropriate level of detail, and clarifies the proposed method and metrics very clearly and as far as I can see also technically correctly, adding also more appropriate imagery and free text description to help the reader to build the appropriate intuition.

The experiments are well chosen and well designed to document the performance and the results are well communicated to allow the reader to pick the most important messages.

While the MGE construction seems relatively simple and the proposed metrics are (I believe) standard in graph analysis, I can see the interest of the community in exploring these for the SNN analysis.

The only weakness is in the description of the topometrics: I presume these are mostly standard in graph analysis. I would like to see this stated more clearly with providing the appropriate references. Also, apart from mentioning the FANMOD algo for Motif, there is not mention of the algorithms to get these graph metrics and their respective complexity. The complexity is mentioned as one of the limitations of the work in section 5 but it is not clear where do these numbers come from - which of the topometrics? I would like to see the connection more clearly.
Together with a missing link to a code-base (is this because of the anonymization of the submission) this reduces the reproducibility of the results and possibility for future re-use. I would strongly encourage the authors to include a link to their code-base in the final version.

A few minor comments are listed in the Requested Changes but these are all fairly cosmetic and should be easy to include into the final version.

---

> ### Author Response · Authors · 2024-03-16
>
> We thank the reviewer for the valuable feedback. Below are our comments about the weaknesses. We also updated the manuscript (the parts related to this reviewer’s comments are in blue) to clarify the following points, and handle all the minor corrections suggested in “Minor change request”. We provide a short reply to some of these minor comments below.
>
> **Complexity is mentioned as one of the limitations of the work in section 5 but it is not clear where do these numbers come from - which of the topometrics?**
>
> Encoding complexity: The complexity in Section 5 refers to the time and space complexity of the encoding of a single BGEs using our proposed unrolled approach. For the whole MGE, the space and time complexity only need to be multiplied by the number of layers (plus the residual connections).
>
>
> **there is not mention of the algorithms to get these graph metrics and their respective complexity. [...] I would like to see the connection more clearly.**
>
> Topometrics’ complexity: The local metrics are easily computed, all together, through a single pass of the edge list (i.e., O(E)) where E is the number of edges in the MGE. For the k-hop neighbor, the complexity is O(E) for k=1 and O(N+E) for k=2. For the Strength Connectivity metrics, the complexity is again O(N+E) for the node-strength (S), while for the k-core decomposition, which is computed using [1], the complexity is O(E). Concerning the Global connectivity metrics, for the connected component, cut-edges and cut-nodes the complexity is   O(N+E) since we used depth-first search (DFS). For the Expansion metrics, we rely on the Implicitly Restarted Lanczos Method [1] (adopted in the standard scipy python library) for computing the eigenvalues, which has a complexity of O(k⋅n⋅m), where n is the size of the matrix, k is the desired number of eigenvalues, and m is the number of Lanczos iterations. We also added a new column in Table 2 to better highlight the complexity of each topometric.
>
> **Together with a missing link to a code-base (is this because of the anonymization of the submission) this reduces the reproducibility of the results and possibility for future re-use. I would strongly encourage the authors to include a link to their code-base in the final version**
>
> Codebase: We will publicly release the codebase of the project. Complementary to our previous response about the topometrics complexity, we would highlight that the codebase implementation has been designed to allow replicating our experiments on limited hardware. Specifically, we implemented our code using Python igraph [1], which wraps a highly optimized C++ code. Of note, almost all metrics can be easily computed in igraph from the standard end-to-end MGE. In fact, we implemented from scratch only the function to compute the Expansion Metrics using an ad hoc Numba (a just-in-time compiler that translates python code to machine code) code. Numba has also been used inside our MGE for the unrolling of convolutional layers, to speed up the iterations over the edge construction (see Eq. 2 and Eq. 3 of the paper).
>
> **Minor**
>
> * Padding: The padding nodes are by default added to L_{i} (i.e., the dimensions of h_i and w_i include the padding nodes).
> r-out and r-in are the absolute counts. The ratio is then used in the regression analysis, as stated in footnote 5.
> * Source nodes cannot appear in the first layer, as sink nodes do not appear in the last one. We added a clarification in the main text.
> * Neighborhood connectivity - are the k-hop neighborhood directional? All the metrics, except for the Expansions (as stated in footnote 3), have been computed in a directional setting.
> * Spectral gap - is this correct? We align our definition to the ones used in [2-4]. In our case. It is indeed the 2nd biggest eigenvalue. However, if for some reason (i.e., due to the rolled encoding), the graph turns out to be d-left-regular, the first and the second eigenvalues have the same values. Hence, we consider the so-called “first non-trivial” eigenvalue.
>
> **References**
>
> [1] Batagelj, Vladimir, and Matjaz Zaversnik. "An o (m) algorithm for cores decomposition of networks." arXiv preprint cs/0310049 (2003).
>
> [2] Pal, Bithika, et al. "A study on the ramanujan graph property of winning lottery tickets." International Conference on Machine Learning. PMLR, 2022.
>
> [3] Hoang, Duc NM, et al. "Revisiting pruning at initialization through the lens of Ramanujan graph." The Eleventh International Conference on Learning Representations. 2022.
>
> [4] Hoang, Duc, et al. "Don’t just prune by magnitude! Your mask topology is a secret weapon." Advances in Neural Information Processing Systems 36 (2023).

---

### Review · Reviewer_782F · 2024-03-11

**Summary Of Contributions:**

This paper proposes a method to predict the performance of sparse neural networks. The proposed method involves a multipartite graph construction of the layers, which allows an end-to-end modeling of a sparse neural network instead of a single layer. The proposed method also analyzes the impact of multiple graph topology metrics that are derived upon the multipartite graph to see how these metrics can predict the performance loss of sparse neural networks.  Experimental results suggest that the proposed MGE graph encoding leads to structural metrics that can better predict SNN performances, while metrics resultant from bipartite graph encoding (with only a single layer) cannot.

**Audience:**

Yes

**Claims And Evidence:**

Yes

**Requested Changes:**

Please see the Weaknesses 1 through 4.

**Strengths And Weaknesses:**

## Strengths
1. This paper challenges a commonly used heuristics of using Ramanujan graph metrics to predict the performance. Results of this paper (Section 4.1) show that Ramanujan graph metrics are not so better than graph density while requiring more computation costs.
2. This paper is generally well written and easy to follow. As a non-expert in sparse neural networks, I can follow the main ideas of this paper easily.
3. Experiments are extensive. The authors test different neural architectures, datasets, and sparse training methods with pretty consistent results.
4. As a non-expert of sparse neural networks, I like the motivation of this work to better predict the performance of an SNN from some general structural metrics. It provide insights about how the sparse training algorithms should be designed.

## Weaknesses
1. It seems that the proposed MGE contains a significantly larger number of nodes than single layer ones. Therefore, I am wondering what is the price we pay for generating such a large graph and calculating these structural metrics. Maybe the authors can shed some lights by giving some computation cost analysis (especially, calculating the global connectivity and expansion of a very large graph takes some time).
2. The authors mention supporting residual layers in Section 3.2. While I can somehow roughly figure out the procedure, it is always better to show how to explicitly include residual layers into the analysis, as residual layers are important components in modern neural networks.
3. Section 4.1 can be enhanced with numeric correlation values. Although visually, these values do show the same trend, it is always better to give some numeric values.
4. In Section 4.2, the authors say that 'compute the pearson correlation coefficient between each topometric and the output of each regressor" to assess the importance of each topometric. I am wondering why not just use the regression coefficient? Also, regarding the section, I am wondering whether there are variances between different datasets (in terms of regression accuracy). If the variance is small, it would better enhance the paper.

---

> ### Author Response · Authors · 2024-03-16
> **Official Comment by Authors - Part I**
>
> We thank the reviewer for the valuable feedback. Below are our comments about the Weaknesses section. We also updated the manuscript (the parts related to this reviewer’s comments are in blue since are shared with reviewer deP5) to better emphasize the following points.
>
> **Maybe the authors can shed some lights by giving some computation cost analysis**
>
> Cost Analysis of topometrics’ complexity: The local metrics are easily computed, all together, through a single pass of the edge list (i.e., O(E)) where E is the number of edges in the MGE. For the k-hop neighbor, the complexity is O(E) for k=1 and O(N+E) for k=2, where N is the number of nodes. For the Strength Connectivity metrics, the complexity is again O(E) for the node-strength (S), while for the k-core decomposition, which is computed using [1], the complexity is O(E). Concerning the Global connectivity metrics, for the connected component, cut-edges and cut-nodes the complexity is   O(N+E) since we used depth-first search (DFS). For the Expansion metrics, the complexity derives from computing the k largest eigenvalues. We rely on the Implicitly Restarted Lanczos Method [1] (adopted in  the standard scipy python library) which has a complexity of O(k⋅n⋅m), where n is the size of the matrix, k is the desired number of eigenvalues, and m is the number of Lanczos iterations. We also added a new column in Table 2 to better highlight the complexity of each topometric.
>
> **The authors mention supporting residual layers in Section 3.2. While I can somehow roughly figure out the procedure, it is always better to show how to explicitly include residual layers into the analysis, as residual layers are important components in modern neural networks**
>
> We included an explanation of how residual connections are included in the proposed MGE in the paper.
>
> **Section 4.1 can be enhanced with numeric correlation values. Although visually, these values do show the same trend, it is always better to give some numeric values.**
>
> Regarding the numerical correlation between Layer-Density and Ramanujan properties, we included in Appendix D the numerical values w.r.t. the three Ramanujan properties over the 7 pruning algorithms, 6 sparsity ratios, and 4 architecture tested.
>
> **In Section 4.2, the authors say that 'compute the pearson correlation coefficient between each topometric and the output of each regressor" to assess the importance of each topometric. I am wondering why not just use the regression coefficient**
>
> We relied on Pearson's correlation for a simple reason: it is always defined in [-1,1], while not all regressors enforce bounds or regularization terms on their coefficients. Note that also using normalization or standardization would not have met this requirement. In other words, using Pearson's correlation allows us to uniformly represent topometric importance across regressors. In turn, this is also crucial to avoid biasing the ranking strategy towards prominent numerical instabilities.
>
> [1] Lehoucq, Richard B., Danny C. Sorensen, and Chao Yang. ARPACK users' guide: solution of large-scale eigenvalue problems with implicitly restarted Arnoldi methods. Society for Industrial and Applied Mathematics, 1998.

---

> > ### Comment · Reviewer_782F · 2024-03-18
> > **Thanks for the response. I have no further questions.**
> >
> > Hello authors,
> >
> > I have read your revision and your response. They are satisfactory in my opinion, and I have no further questions regarding the submission. Thanks.

---

> ### Author Response · Authors · 2024-03-16
> **Official Comment by Authors - Part II**
>
> **Also, regarding the section, I am wondering whether there are variances between different datasets (in terms of regression accuracy). If the variance is small, it would better enhance the paper.**
>
> The main idea behind our regression analysis is to be as dataset-independent as possible. Hence, we performed the regression study on the topometrics w.r.t. the performance drops aggregated across three different classification datasets of increasing complexity (namely, CIFAR-10, CIFAR-100, and Tiny-Imagent). The main reason for such a choice is to provide (as shown in the section on the ranker by topometric mixture) generalizable topometric importance.. We hope that this approach -being indeed dataset-independent- would encourage other researchers to build on our work to study other SNNs.
> As suggested by the reviewer, we performed this regression analysis separated by dataset. The numerical results are shown below. Please note that the number of sparse structures in each analysis is one-third compared to the number of structures in the dataset-independent analysis reported in the paper.
> In this new analysis (that we include only in this rebuttal reply, for the reviewer’s convenience) the results turn out to follow the same trend shown in the paper: the model-fixed setting is the one where higher values of Adjusted-R^2 are reached, while for the sparsity-fixed case, the Adjusted-R^2 coefficient increases when the sparsity ratio increases since the difference in performance drop among PaI algorithms is larger (hence easy to predict) at higher sparsity values. Furthermore, the variance across datasets turns out to be low in the model-fixed case, and moderate in the sparsity-fixed setting. The reason for this is likely the lower number of data points in this new regression analysis compared to the one reported in the paper, as well as the fact that the performance drop is highly dependent on the combination of sparsity ratio and dataset (see Table 9-10-11 in Appendix B).
>
> The regression setting has been kept fixed as in the paper.
>
> | Model | CIFAR-10 \( R^2 \) | CIFAR-100 \( R^2 \) | TinyImagenet \( R^2 \) | Mean ± Std |
> |--------------|-------------------|---------------------|-----------------------|------------------------------------------|
> | s = 0.6 | 0.542 ± 0.061 | 0.286 ± 0.047 | 0.732 ± 0.032 | 0.52 ± 0.182 |
> | s = 0.8 | 0.620 ± 0.094 | 0.548 ± 0.037 | 0.845 ± 0.024 | 0.671 ± 0.127 |
> | s = 0.9 | 0.894 ± 0.019 | 0.783 ± 0.061 | 0.808 ± 0.053 | 0.828 ± 0.048 |
> | s = 0.95 | 0.839 ± 0.017 | 0.851 ± 0.008 | 0.807 ± 0.039 | 0.832 ± 0.019 |
> | s = 0.98 | 0.906 ± 0.011 | 0.615 ± 0.031 | 0.554 ± 0.067 | 0.692 ± 0.154 |
> | CONV-6 | 0.979 ± 0.003 | 0.923 ± 0.017 | 0.885 ± 0.019 | 0.929 ± 0.039 |
> | Resnet20 | 0.900 ± 0.017 | 0.970 ± 0.004 | 0.924 ± 0.013 | 0.931 ± 0.029 |
> | Resnet-32 | 0.876 ± 0.016 | 0.884 ± 0.013 | 0.950 ± 0.003 | 0.904 ± 0.033 |
> | Wide-Resnet-28-2 | 0.974 ± 0.003 | 0.985 ± 0.002 | 0.967 ± 0.002 | 0.976 ± 0.007 |

---

### Decision · Action_Editor_CYhq · 2024-04-19

**Recommendation:** Accept as is

**Comment:**

The reviewers appreciated the direction of using topological metrics to better understand and predict the performance of sparse neural networks (SNNs), as well as the proposed MGE methodology and the extensive experiments conducted. They listed as strengths the fact that the paper is challenging common metrics, and single-layer approaches; it is proposing a simple but effective idea with MGE.

At the same time, they highlighted a number of issues, mostly dealing with presentation. Specifically, they asked to revise the description of the topological metrics to make them more clear for the TMLR audience, and list the complexity of the discussed approaches. Other criticisms regarded missing details from the experiments and on how to interpret results in tables.

Authors exhaustively answered all the concerns during the rebuttal and provided a revised version of the manuscript that satisfied all the reviewers. As such, the paper is accepted as is.

**Audience:**

Sparsifying neural networks is a hot topic in ML, and understanding what happens to sparse networks and predicting their performance is definitely interesting for the TMLR audience.

**Claims And Evidence:**

This work traces a connection between the performance of sparsifying schemes of neural networks via pruning with metrics in topology of graphs. Specifically, authors propose to use a multipartite graph representation called MGE for sparse neural nets that leverages topological metrics, is input aware and allows to better predict the performance drop. Further analysis on combinations of metrics highlights which are most important and suggests a way to aggregate them to accurately predict performance of sparse neural nets. Experiments support the aforementioned claims.